# Iridium metallene oxide for acidic oxygen evolution catalysis

Qian Dang[1,2,3,6], Haiping Lin[2,6], Zhenglong Fan[2,6], Lu Ma[4,6], Qi Shao[1✉], Yujin Ji[2], Fangfang Zheng[2], Shize Geng[1,2], Shi-Ze Yang[5✉], Ningning Kong[2], Wenxiang Zhu[2], Youyong Li[2✉], Fan Liao[2], Xiaoqing Huang[3✉] & Mingwang Shao[2✉]

Exploring new materials is essential in the field of material science. Especially, searching for optimal materials with utmost atomic utilization, ideal activities and desirable stability for catalytic applications requires smart design of materials' structures. Herein, we report iridium metallene oxide: 1T phase-iridium dioxide ($IrO_2$) by a synthetic strategy combining mechanochemistry and thermal treatment in a strong alkaline medium. This material demonstrates high activity for oxygen evolution reaction with a low overpotential of 197 millivolt in acidic electrolyte at 10 milliamperes per geometric square centimeter (mA $cm_{geo}^{-2}$). Together, it achieves high turnover frequencies of 4.2 $s_{UPD}^{-1}$ (3.0 $s_{BET}^{-1}$) at 1.50 V vs. reversible hydrogen electrode. Furthermore, 1T-$IrO_2$ also shows little degradation after 126 hours chronopotentiometry measurement under the high current density of 250 mA $cm_{geo}^{-2}$ in proton exchange membrane device. Theoretical calculations reveal that the active site of Ir in 1T-$IrO_2$ provides an optimal free energy uphill in *OH formation, leading to the enhanced performance. The discovery of this 1T-metallene oxide material will provide new opportunities for catalysis and other applications.

[1] College of Chemistry, Chemical Engineering and Materials Science, Soochow University, 215123 Jiangsu, P. R. China. [2] Institute of Functional Nano & Soft Materials (FUNSOM), Soochow University, 215123 Jiangsu, P. R. China. [3] State Key Laboratory of Physical Chemistry of Solid Surfaces, College of Chemistry and Chemical Engineering, Xiamen University, 361005 Xiamen, P. R. China. [4] NSLS-II, Brookhaven National Laboratory, Upton, NY 11973, USA. [5] Eyring Materials Center, Arizona State University, Tempe, AZ 85287, USA. [6] These authors contributed equally: Qian Dang, Haiping Lin, Zhenglong Fan, Lu Ma. ✉email: qshao@suda.edu.cn; shize.yang@asu.edu; yyli@suda.edu.cn; hxq006@xmu.edu.cn; mwshao@suda.edu.cn

The increasing worldwide energy and pollution crisis demand the highly efficient production of renewable energy[1–6]. Electrochemical water splitting provides a promising way to make efficient energy conversion applicable, yet its biggest challenge comes from the slow kinetics of oxygen evolution reaction (OER) at the anode and the harsh working condition in the strong acidic environment[7–10]. More importantly, it is required for developing OER catalysts in acidic pH regime for their direct impact on making the proton exchange membrane (PEM) viable on an industrial scale[11–13].

Iridium oxide ($IrO_2$) is currently the only material that can remain stable under the harsh acidic OER condition, while its progress has been largely impeded by its unsatisfied activities and ultralow-earth abundance[14–16]. Several strategies have been reported to develop complex iridium-based oxides, such as: surface restructuration strategy[15], amorphous strategy[17], and doping strategy[16], yet still leaving a large room for improvement. Therefore, designing $IrO_2$ catalysts with exceptional activity and stability and high atomic utilization becomes a central issue for breaking through this trade-off.

Searching for an $IrO_2$ material in low dimensionality provides a promising strategy to overcome this dilemma. The allure of searching an advanced structure can be mapped to its increased possibilities that could revolutionize the oxygen evolution catalysis due to their unique atom arrangement, bonding geometry, electronic structure, and unexpected surface surroundings[18–22]. In addition, two-dimensional (2D) materials are highly desirable for surface-sensitive reactions in terms of cost-effectiveness due to their near-100% atom utilization, which provides the maximized density of surface active sites[23–27]. It also provides an ideal platform for catalytic optimization by virtue of favorable mass transport, fascinating mechanical behaviors, and tunable electronic properties[28–30]. As the surface energy increases with thinning the crystal thickness[31,32], the stability of phase increases with decreasing the layer thickness. Such beneficial features may introduce insights into the role of dimensionality engineering in discovering advanced materials.

In terms of the synthetic method for discovering various materials, mechanochemical synthesis provides a unique means for the pursuit of metastable polymorphous products[33] and the strong alkaline medium is feasible for the exploration of metastable compounds[34]. Hence, to obtain an advanced material with high degree of crystallinity, a combination of mechanochemistry, strong alkaline environment and thermal treatment may be a reasonable attempt. However, it is a challenge to synthesize under these harsh preparation conditions. Motivated by all these

possibilities, we report an iridium metallene oxide: 1 T-phase-$IrO_2$ using a robust synthetic route referred as the "mechano-thermal" method in a strong alkaline medium. The structure of this material is characterized by a space group: P-3m1 (164) ($a = b = 3.11$ Å and $c = 6.91$ Å). $1T\text{-}IrO_2$ displays a substantially low overpotential of 197 millivolt (mV) at 10 milliamperes per geometric square centimeter (mA $cm_{geo}^{-2}$) for acidic OER, with the combination effect of the optimized Ir reaction site, high number of active sites and atomic utilization. Notably high turnover frequencies (TOFs) of 4.2 $s_{UPD}^{-1}$ (3.0 $s_{BET}^{-1}$) were obtained by $1T\text{-}IrO_2$. More importantly, it also exhibits little activity decline for 126 h continuous test under the high current of 250 mA $cm_{geo}^{-2}$ in PEM device. Density functional theory (DFT) calculations reveal that the binding of hydroxyl groups on the $1T\text{-}IrO_2$ is an endothermic process with an optimized free energy uphill closing to the equilibrium potential of 1.23 eV, which is in sharp contrast with the $IrO_2$ of rutile phase. As a result, the typical potential-limiting step of subsequent *OH dissociation or *O formation in OER is significantly promoted.

## Results

**Morphological characterizations of $1T\text{-}IrO_2$.** The synthetic conditions of $1T\text{-}IrO_2$ are challenging: high temperature (800 °C), a strong alkaline environment and the continuous grinding force. In this work, we successfully realized all the strict requirements in our "home-made" reactor, where the main body is made by the high-quality corundum, as the suitable material that can remain stable in an harsh environment: high temperature, continuously grinding and strong alkaline (Fig. 1 and Supplementary Fig. 1). Iridium trichloride ($IrCl_3$) and potassium hydroxide (KOH) were chosen as the reaction precursors. The mixture was grinded with the temperature increasing from room temperature to 800 °C with the heating rate of 10 °C min$^{-1}$ and kept at 800 °C for 3 h in the mechano-thermal reactor. The detailed synthesis process can be found in the Methods section. The obtained $1T\text{-}IrO_2$ powder shows the dark blue color and its water dispersion in a cuvette shows the light blue color, which are totally different from those of $IrO_2$ with rutile phase (Supplementary Fig. 2a, b).

Scanning electronic microscopy (SEM) image reveals that the 2D nanosheet is the dominant product (Fig. 2a). Transmission electron microscopy (TEM) images are further used to give detailed morphology information, where the thin-sheet structure can be observed (Fig. 2b and Supplementary Fig. 2c, d). Atomic force microscopy (AFM) analyses also confirm the ultrathin 2D structure of $1T\text{-}IrO_2$, with the thickness of 3–5 nm (Supplementary Fig. 3). The chemical composition was then verified by TEM energy-dispersive X-ray spectroscopy (TEM-EDX), where the atomic ratio of Ir and O is about 1: 2 (Supplementary Fig. 2e, f). Other evidence such as the cross-sectional view showing monolayer and bilayer regions, and linear annular dark-field scanning transmission electron microscopy (STEM-ADF) intensity is also given in Supplementary Fig. 2g. Elemental distributions of Ir and O were then detected by a STEM-EDX mapping, where Ir (red) and O (green) uniformly distribute through the whole nanosheet (Supplementary Fig. 2h).

**Analysis of $1T\text{-}IrO_2$ structure.** The X-ray diffraction (XRD) technique was then performed to reveal the crystal structure of $1T\text{-}IrO_2$ nanosheets using Cu Kα radiation ($\lambda = 1.5406$ Å). As shown in Fig. 2d, sharp Bragg diffraction peaks confirmed the high crystallinity and no diffraction peaks that belong to the starting materials or the rutile-$IrO_2$ [Joint Committee on Powder Diffraction Standards (JCPDS) No. 88-0288] are detected. The peaks correspond to the (0001) and (0002) diffraction planes, respectively, indicating the layered structure along the c-axis. The

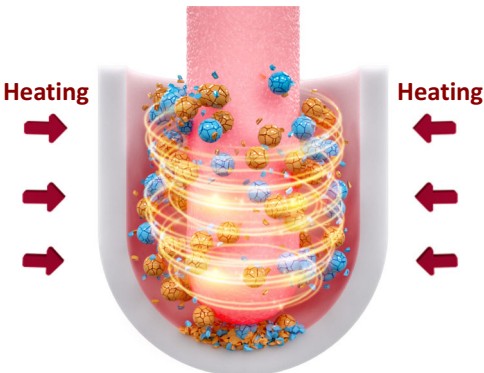

**Fig. 1 Schematic representation of the mechano-thermal reactor for preparing $1T\text{-}IrO_2$, where the mechanical and thermal operations are controlled simultaneously.** The blue and yellow balls indicate $IrCl_3$ and KOH, respectively.

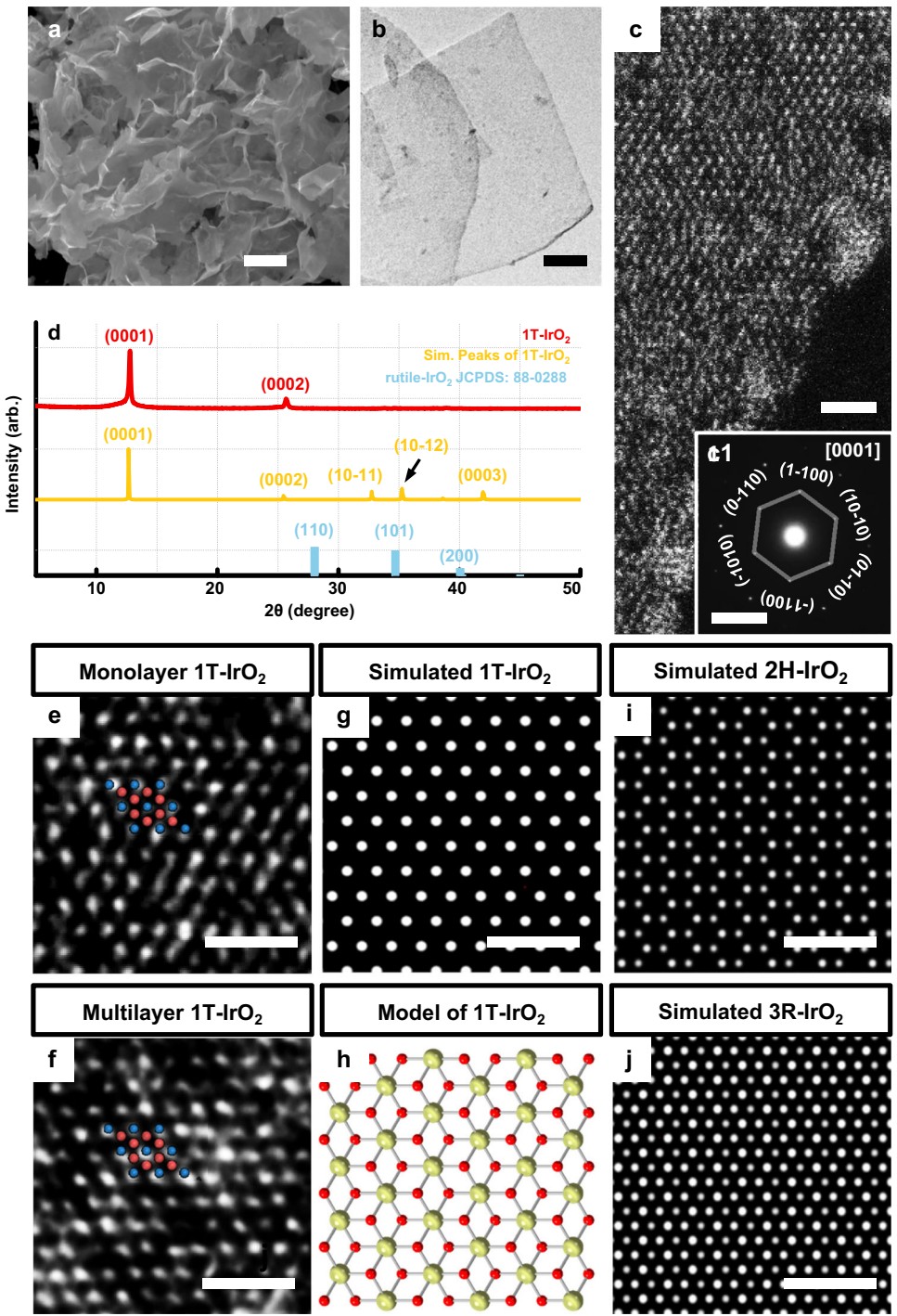

**Fig. 2 Structural characterizations of 1T-IrO₂. a** The SEM and **b** TEM images of 1T-IrO₂, representing ultrathin morphology. **c** The aberration-corrected HAADF-STEM image of monolayer 1T-IrO₂. The hexagonal pattern is indicative of the [0001] projection. **c1** The SAED pattern of 1T-IrO₂ in Fig. 2c, showing the six-fold rotational symmetry. **d** Comparison of XRD patterns of 1T-IrO₂ (red curve), simulation X-ray diffraction peak (yellow curve) of 1T-IrO₂ and the rutile-IrO₂ (blue line) (JCPDS No. 88-0288). The XRD measurement showing high orientation of the layered structure of 1T-IrO₂ along the c-axis. **e, f** HAADF-STEM images of the monolayer and multilayer 1T-IrO₂, which show identical atom arrangement. Ir and O are represented by blue and red spheres. **g, i, j** Simulated TEM images of IrO₂ with 1 T, 2H and 3 R phases. Our result of 1T-IrO₂ is in accordance with the AA stacking, different from those shown by AB stacking (2H) and ABC stacking (3 R). **h** The atom model of 1T-IrO₂, where the yellow and red balls are Ir and O. The scale bars in **a, b, c, and c1** are 2 μm, 100 nm, 1.25 nm, and 5 1/nm, respectively. The scale bars in **e, f, g, i, j** are 1 nm.

peak at about 12.81° is an important fingerprint to distinguish the unit-cell parameter $c$ along $c$ direction, which is calculated to be ~6.91 Å.

The aberration-corrected STEM-ADF imaging technique was further used to clarify the phase of 1T-IrO₂[35,36]. A typical STEM-ADF image from monolayer region is shown in Fig. 2c with the edge along the (10–10) surface. Since the layered flake is very sensitive to the electron beam, a very small current of 23 pA at 200 kV was used to reduce beam damage. The selective-area electron diffraction (SAED) confirms its crystalline nature

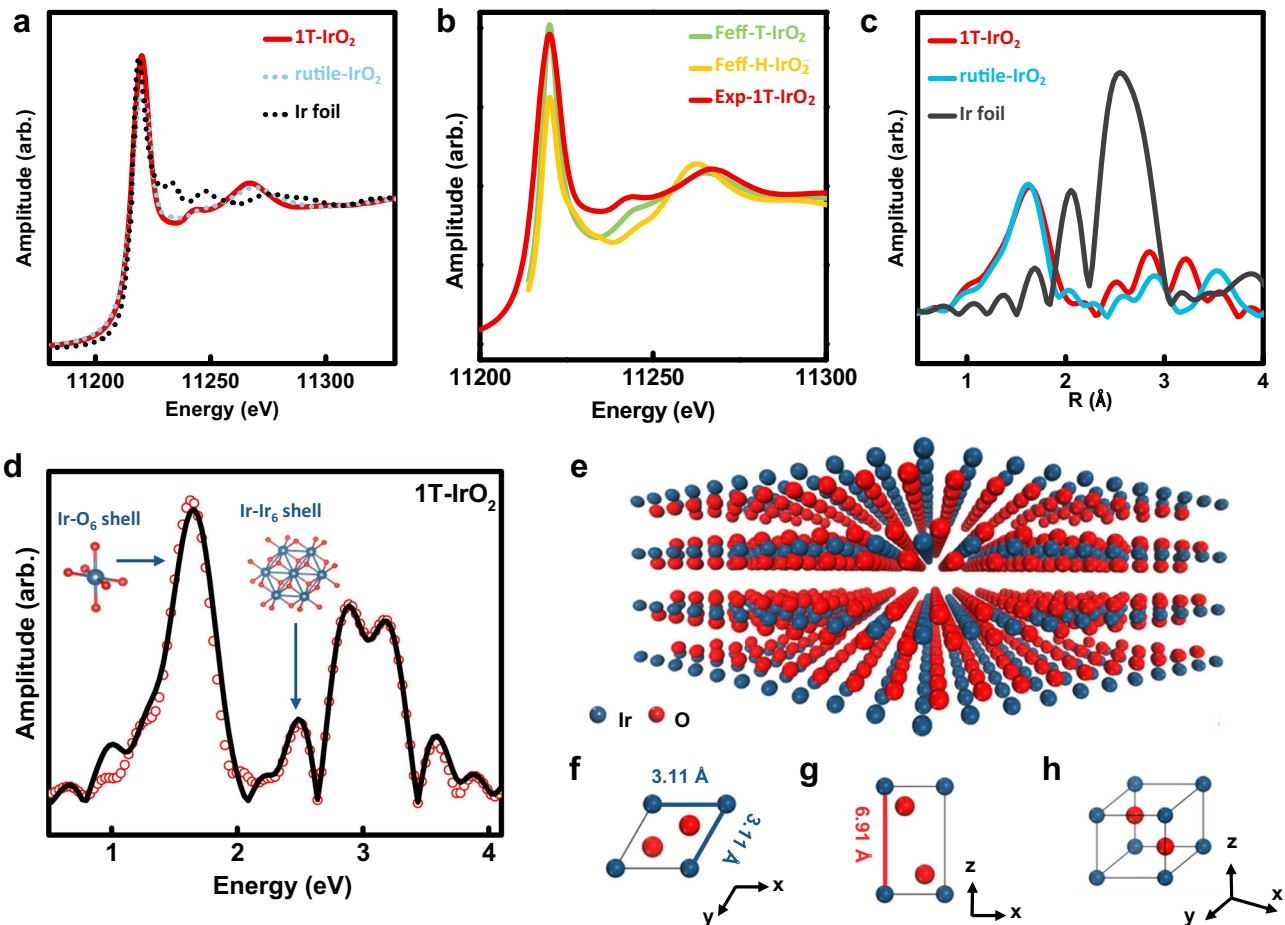

**Fig. 3 Structure representations of 1T-IrO₂.** **a** Ir-LIII edge XANES of 1T-IrO₂, rutile-IrO₂, and Ir foil. **b** Ir-LIII edge XANES of T-phase IrO₂ and H-phase IrO₂ are simulated using Feff9 code and compared with experimental result of 1T-IrO₂. **c** $k^2$-Normalized Ir-LIII edge EXAFS of 1T-IrO₂, rutile-IrO₂, and Ir foil. **d** $k^2$-Normalized Ir-LIII edge EXAFS of 1T-IrO₂, where the Ir–O₆ and Ir–Ir₆ shells are highlighted. **e** Atomic structure of layered 1T-IrO₂. **f–h** Schematic of the unit cell of 1T-IrO₂ with the lattice constants of **a** = **b** = 3.11 Å and **c** = 6.91 Å. The refined structure models are obtained based on the XRD and the aberration-corrected HAADF-STEM analysis. Ir and O are represented by blue and red spheres, respectively.

(Fig. 2c1). As the intensity of atomic columns is proportional to the atomic number, the hexagonal array of Ir atoms is determined while the intensity of oxygen columns is too weak to be seen[37]. Then there are only two questions about the crystal structure of this material: one is the layer stacking and the other is the atomic coordination environment surrounding Ir atoms. Figure 2e, f gives the information of the typical atomic-resolution images of both the monolayer and multilayer 1T-IrO₂, which show the identical appearance. Comparing with the simulated STEM-ADF images of AB stacking and ABC stacking, it unambiguously suggests that our 1T-IrO₂ is AA stacked (Fig. 2g, h), excluding the 2H (AB stacking) and 3 R (ABC stacking) crystal arrangements (Fig. 2i, j).

The other important information is the coordination information between Ir and O atoms. We addressed this question by two different approaches. First, according to the literature there are two possible structures for oxygen coordination: one is the trigonal (T) phase and the other is the hexagonal (H) phase[36]. We adopted the H and T-phase from MoS₂[37] and relaxed the structures using DFT. The H-phase structure of IrO₂ collapsed upon structure relaxation while the T-phase of IrO₂ keeps stable, suggesting that the T-phase is more energetic-favorable at ground state. We further carried out synchrotron X-ray absorption using the Ir-LIII edge to probe the coordination environment[38]. The X-ray absorption near-edge spectroscopy (XANES) results for 1T-IrO₂, rutile-IrO₂ and Ir metal foil are shown in Fig. 3a. To

further distinguish the oxygen coordination environment between H and T phases, we simulated the XANES curved using Feff9 code as shown in Fig. 3b[39]. The better agreement between experimental 1T-IrO₂ and the T-phase further confirms the trigonal coordination environment. The extended X-ray absorption fine structure (EXAFS) results were also collected and Fourier transformed to show the real space coordination environments as shown in Fig. 3c for 1T-IrO₂, rutile-IrO₂, and Ir metal foil. The spectrum for 1T-IrO₂ is fitted and shown in Fig. 3d with the two coordination shells highlighted for the Ir–O₆ shell and Ir–Ir₆ shell. The coordination number and bond lengths for these two shells are fitted to be 6.0/2.00 Å and 6.0/3.12 Å.

We thus propose a structure model as shown in Fig. 3e–h. The calculated XRD pattern derived from the proposed 1T-IrO₂ model is well consistent with the reflections observed in the experimental XRD pattern (Fig. 2d). According to the proposed model, 1T-IrO₂ crystallizes in the space group of P-3m1, which is totally different from that of rutile-IrO₂ (space group: P4₂/mnm). The corresponding schematic of layered 1T-IrO₂ is shown in Fig. 3e to help vividly visualize the atomic structure. All the lattice parameters are given in Supplementary Table 1.

**The synthesis of 1T-IrO₂.** The direct synthesis of 1 T-phase is full of challenges since the formation energy of 1 T-phase is higher than that of the rutile phase, therefore making it important to

develop a rational synthetic strategy that leads to this phase. In this work, the mechano-thermal reactor plays a key role in obtaining this phase of $IrO_2$ since it provides a powerful way to activate the starting materials and obtain the metastable polymorphous forms[33,34]. Only the rutile-$IrO_2$ with low crystalline was obtained when annealing the mixture of $IrCl_3$ and KOH prepared by a simple mixing process (Supplementary Fig. 4a). Moreover, KOH is also essential for the formation of 1T-$IrO_2$, where only $IrO_2$ nanoparticles with the rutile phase were obtained with the absence of KOH (Supplementary Fig. 4b, d–g). This indicates that the strong alkaline environment provide a critical environment for the growth of the oxide materials[40,41]. In addition, the thermal treatment is vital for the formation of high degree crystalline crystal. We checked the XRD pattern of the product annealing at 200 °C without double-distilled water washing, where the reaction intermediates of $K_{0.25}IrO_2$ and $K_4IrO_4$ can be detected (Supplementary Fig. 5). However, no trace of K was detected in 1T-$IrO_2$ (Supplementary Fig. 6). Only the amorphous product can be obtained when the annealing temperature of mechano-thermal process is 400 °C (Supplementary Fig. 4c). With increasing annealing temperature from 500 °C to 800 °C, 1T-$IrO_2$ becomes highly crystalline with exhibiting strong *c*-axis preferable growth and the clear 2D sheet morphology (Supplementary Fig. 4h–k). Therefore the combination of mechanical processing and annealing treatment in a strong alkaline medium is crucial for the formation of 1T-$IrO_2$. We further evaluated the phase structure of 1T-$IrO_2$ annealed at high temperature (900 °C) by analyzing the X-ray diffraction result. The thermodynamically stable rutile phase was detected, further indicating the metastable property of 1T-$IrO_2$ (Supplementary Fig. 7a, b).

To gain a better view of intrinsic structure of 1T-$IrO_2$, X-ray photoelectron spectroscopy (XPS) was used to reveal the electronic states of 1T-$IrO_2$. Based on the high-resolution XPS spectra in Supplementary Fig. 7c, the O 1*s* peaks from Ir–O binding, are clearly observed in both 1T-$IrO_2$ and rutile-$IrO_2$. More importantly, the Ir 4*f* binding energies of 1T-$IrO_2$ are shifted to the higher binding energy than those of rutile-$IrO_2$ (Supplementary Fig. 7d), suggesting that majority of $Ir^{3+}$ species exist on the surface of 1T-$IrO_2$[42]. Therefore the surface Ir in 1T-$IrO_2$ is in a more metallic state. The contrary shifting of XPS peak towards high energy with lower valence state might be due to the strong electrons correlation effect in Ir. This result is also in good agreement with those measured from the synchrotron XANES result shown in Fig. 3a and Supplementary Fig. 7e, f.

**OER performance of 1T-$IrO_2$.** Since $IrO_2$ is regarded as the most promising catalyst for acidic OER, the OER performance of 1T-$IrO_2$ deserves deep evaluation[15–17]. The OER performances of $IrO_2$ with rutile phase (Rutile-$IrO_2$) and the two reference electrocatalysts: C-$IrO_2$ and commercial-Ir/C (C-Ir/C) were also measured for comparison. The OER tests were measured using a standard three-electrode system in 0.1 M $O_2$-saturated $HClO_4$ electrolyte. The reference electrode was calibrated before OER tests (Supplementary Fig. 8). As revealed in Fig. 4a, the linear sweep voltammetry (LSV) shows that 1T-$IrO_2$ has the most active polarization curve, with an ultralow overpotential of 197 mV to reach the current density of 10 mA $cm_{geo}^{-2}$. This overpotential is 100 mV lower than that of Rutile-$IrO_2$, suggesting the important role of the phase in tuning the electrochemical performance. For comparison, higher overpotentials were obtained by C-$IrO_2$ and C-Ir/C, with delivering 276 mV and 311 mV, respectively (Fig. 4a). Tafel slope is another important parameter to evaluate the OER performance. The lowest Tafel slope of 49 mV $dec^{-1}$ was obtained by 1T-$IrO_2$, compared to those measured by Rutile-$IrO_2$,

C-$IrO_2$, and C-Ir/C (Fig. 4b), suggesting that the surface environment of 1T-$IrO_2$ is more favorable for OER pathway. We further compared the geometrical current densities at 1.50 V vs. reversible hydrogen electrode (RHE) of different catalysts. 1T-$IrO_2$ exhibit the highest geometrical current density of 52.7 mA $cm_{geo}^{-2}$, which is 9.6, 6.3 and 10.8 times high than those of Rutile-$IrO_2$ (5.5 mA $cm_{geo}^{-2}$), C-$IrO_2$ (8.4 mA $cm_{geo}^{-2}$), and C-Ir/C (4.9 mA $cm_{geo}^{-2}$) (Fig. 4c).

The intrinsic activity of 1T-$IrO_2$ was further evaluated by determining the mass activity and TOFs at 1.50 V vs. RHE. The mass activities of all the catalysts are calculated by normalizing the mass of iridium and the TOF values of all the catalysts are calculated by normalizing the mercury underpotential deposition (UPD)-based surface area, Brunauer-Emmett-Teller (BET) surface area or electrochemically active surface areas (ECSAs)-based surface area. The methods to determine the UPD, BET or ECSA-based surface areas of different catalysts were collected in Methods section (Supplementary Figs. 9–12, Supplementary Tables 2–6 and Supplementary Note 1). As shown in Fig. 4d, 1T-$IrO_2$ exhibits the highest mass activity of 296.8 mA $mg_{Ir}^{-1}$, which is 9.7 times higher than that of Rutile-$IrO_2$. More importantly, we calculated the TOF values of different catalysts (Fig. 4e, f and Supplementary Fig. 13). The TOF values follow the order in the sequences of 1T-$IrO_2$, C-$IrO_2$, Rutile-$IrO_2$ and C-Ir/C with 1T-$IrO_2$ exhibiting the highest TOF value of 4.2 $s_{UPD}^{-1}$ (3.0 $s_{BET}^{-1}$ and 2.7 $s_{ECSA}^{-1}$) at 1.50 V vs. RHE. We further evaluated the OER performances of 1T-$IrO_2$ in the scale-up experiment performed with the $1 \times 1$ $cm^2$ electrodes (Methods section). The mass and TOF values of 1T-$IrO_2$ are higher than those of C-$IrO_2$, Rutile-$IrO_2$ and C-Ir/C (Supplementary Fig. 14).

We also evaluated the OER performance of 1T-$IrO_2$ with using 0.5 M $H_2SO_4$ as the electrolyte. 1T-$IrO_2$ shows the lowest overpotential of 235 mV at the current density of 10 mA $cm_{geo}^{-2}$, compared to 319, 302, and 340 mV of Rutile-$IrO_2$, C-$IrO_2$, and C-Ir/C (Supplementary Fig. 15a). The results are reasonable since the OER activities of Ir based catalysts in $HClO_4$ are better than in $H_2SO_4$[43]. The intrinsic activity of 1T-$IrO_2$ in 0.5 M $H_2SO_4$ was also evaluated. 1T-$IrO_2$ exhibit the TOF values of 2.0 $s_{UPD}^{-1}$ (1.5 $s_{BET}^{-1}$ and 1.3 $s_{ECSA}^{-1}$) (Supplementary Fig. 15d–f and Supplementary Tables 7, 8). After carefully evaluating those state-of-the-art catalysts previously reported, we place 1T-$IrO_2$ among the most efficient catalysts for OER (Supplementary Table 8).

The operating stability is the key issue when considering the implementation for the practical application since most metal oxides are prone to dramatically activity loss under acidic OER condition. Therefore more reliable data collection is performed, based on the standard measurements proposed by Kibsgaard and Chorkendorff[44]. As shown in Fig. 4g, 1T-$IrO_2$ remained 98% of origin activity under the high current density of 50 mA $cm_{geo}^{-2}$ after a 45 h stability test. However, Rutile-$IrO_2$, C-$IrO_2$, and C-Ir/C totally deactivated after the long-term test under the high current density (Fig. 4g and Supplementary Fig. 16). 1T-$IrO_2$ also represents much enhanced stability compared to other catalysts previously reported (Supplementary Table 9). In addition, inductively coupled plasma source mass spectrometer (ICP-MS) test was carried out to check the concentrations of dissolved Ir in 1T-$IrO_2$ and C-$IrO_2$ after the prolonged stability test under a constant voltage of 1.53 V vs. RHE (Supplementary Fig. 17a). As shown in Supplementary Fig. 17b, the limited concentration of dissolved Ir ions for 1T-$IrO_2$ after 45 h stability test is detected, suggesting its remarkable OER stability. We also calculated the s-number to reveal stability property of 1T-$IrO_2$ (Supplementary Fig. 18). Form the data in Supplementary Table 10, all the s-numbers at different working potentials are greater than $1 \times 10^6$, larger than the s-number of commercial $IrO_x$ catalyst ($5 \times 10^5$) in acidic media[45]. The post-reaction UPD-based surface

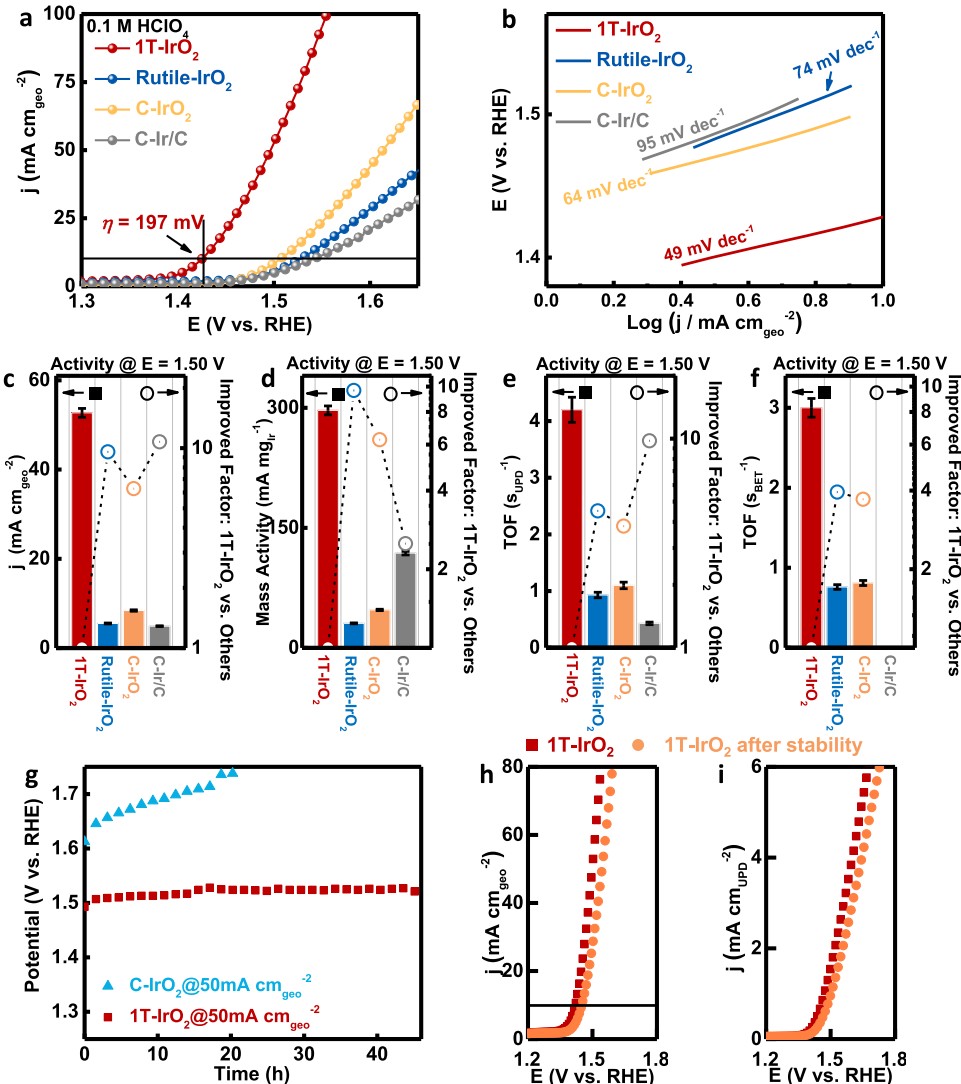

**Fig. 4 OER performance of 1T-IrO$_2$. a** Polarization curves of 1T-IrO$_2$, Rutile-IrO$_2$, and commercial catalysts (C-IrO$_2$ and C-Ir/C) in O$_2$-saturated 0.1 M HClO$_4$ electrolyte with iR-correction, where 1T-IrO$_2$ deliver an overpotential of 197 mV to achieve the current density of 10 mA cm$_{geo}^{-2}$. **b** Tafel plots obtained from the polarization curves in a. **c–f** Comparison of geometric activities, mass activities, UPD-based TOF, and BET-based TOF values at 1.50 V vs. RHE for 1T-IrO$_2$, Rutile-IrO$_2$, C-IrO$_2$, and C-Ir/C. **g** Chronopotentiometry performance under a high constant current density of 50 mA cm$_{geo}^{-2}$ for 1T-IrO$_2$ and C-IrO$_2$, where 1T-IrO$_2$ maintain high electrochemical activity. **h** The geometric polarization curves and **i** UPD-based polarization curves of 1T-IrO$_2$ before and after stability test under the high current density of 50 mA cm$_{geo}^{-2}$, where the change of UPD-based active area-based activity in 1T-IrO$_2$ is very weak. Error bars are means ± SD ($n = 3$ replicates).

area of 1T-IrO$_2$ was also evaluated (Supplementary Fig. 19a). The decline of active area-based activity in 1T-IrO$_2$ is very small, suggesting that the loss of OER activity is largely due to the drop of catalyst from the electrode during the stability test (Fig. 4i). TEM image shows that the morphology is largely maintained after the long-term stability test under the high current density of 50 mA cm$_{geo}^{-2}$. TEM-EDX mapping confirms the uniform distribution of Ir and O in 1T-IrO$_2$ after the stability test (Supplementary Fig. 19b, c).

More importantly, in a proof-of-principle of its practical use, we examined the catalytic performance of 1T-IrO$_2$ in PEM device. 1T-IrO$_2$ reveals the high current density of 253 mA cm$_{geo}^{-2}$ at the voltage of 1.50 V, higher than 14.1 mA cm$_{geo}^{-2}$ of C-IrO$_2$ (Supplementary Figs. 20, 21), which places 1T-IrO$_2$ among the most efficient catalysts reported (Supplementary Table 11). It also operates stably for 126 h at the high current of 250 mA cm$_{geo}^{-2}$ in PEM device with the Faradic efficiencies approaching to 100% during the long-term stability test, indicating its promising

industrial application (Supplementary Figs. 22–24 and Supplementary Table 12). As shown in Supplementary Fig. 25, 1T-IrO$_2$ shows the high stability for the start-stop cycles test. The sheet morphology and crystal structure of 1T-IrO$_2$ also largely remain after the stability test in PEM device (Supplementary Fig. 26). Based on the XRD result, the typical (0001) diffraction peak at 12.81° was detected and no signal from the rutile phase could be observed, suggesting the high stability of 1T-IrO$_2$ in electrochemical catalysis. SAED image shows that 1T-IrO$_2$ remain its high crystalline nature (Supplementary Fig. 26a1). TEM-EDX mapping confirms the uniform distribution of Ir and O in 1T-IrO$_2$ after the stability test (Supplementary Fig. 26b). According to Supplementary Fig. 26c, the regular array of Ir atoms can be detected. No amorphous phases were detected, indicating the high stability of 1T-IrO$_2$. XPS analysis reveals that Ir and O elements are clearly detected (Supplementary Fig. 26e, f). X-ray absorption near-edge structure (XANES) of 1T-IrO$_2$ after the stability test was carried out. As shown in Supplementary Fig. 26g, the valence state

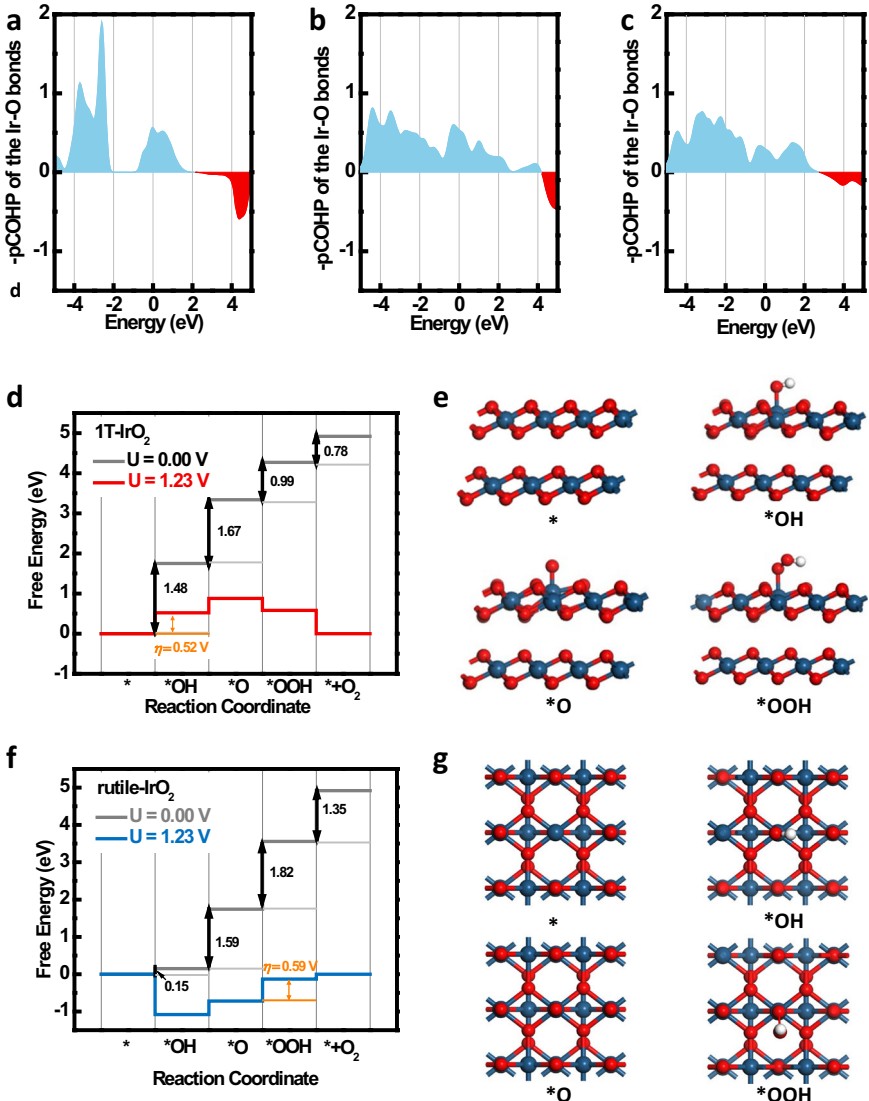

**Fig. 5 OER mechanism of 1T-IrO₂.** The pCOHP plots of **a** the Ir–O bond in 1T-IrO₂, **b** the shorter Ir–O bond in rutile-IrO₂, and **c** the longer Ir–O bond in the rutile-IrO₂. The energies have been aligned to the highest occupied states. The blue and red contours represent the bonding and antibonding contributions in Ir–O interactions, respectively. The thermal stability of rutile phase should be better than the 1T-phase, indicating that 1T-phase is a metastable phase. **d, e** The free energy profile of OER over the 1T-IrO₂. **f, g** The free energy profile of OER over rutile-IrO₂ (110) surface. The Ir, O, and H atoms are represented with the blue, red, and white circles, respectively.

of 1T-IrO₂ does not change much compared to that of the reference rutile phase IrO₂, suggesting its high stability. All these results conclude that the 1T-phase can be largely maintained under the acidic OER condition.

**DFT simulations of 1T-IrO₂ as an OER catalyst**. DFT calculations have been conducted to understand the intrinsic bonding characteristics and catalytic activity in OER. Please note that there are two types of Ir–O bonds in the rutile phase (1.93 Å and 2.02 Å), while only one kind of Ir–O bond in the 1T-phase. The relative stability of the 1T and rutile phases can be qualitatively evaluated by the projected crystal orbital Hamilton population (pCOHP) of the three types of Ir–O bonds. The COHP plots can be employed to analyze the contribution of bonding and antibonding states in the density of states. When chemical bonds are formed in a material, the bonding orbitals are typically occupied, while the antibonding orbitals are often unoccupied. If the unoccupied antibonding orbitals are very close to the Fermi

energy, they become occupied by the excited electrons at elevated temperatures due thermal excitations. Hence, this material may probably decompose at high temperatures. However, if the unoccupied antibonding orbitals are well above the Fermi energy, this suggests that the pumping of electrons into these states will require high thermal excitations. Correspondingly, the material is expected to possess excellent thermal stability. As shown in Fig. 5a–c, in all three Ir–O bonds, the energy range below 2.0 eV are dominated by the bonding contributions, implying that both 1T and rutile phases are stable structures at mild conditions. The antibonding contributions, however, appear from 2.1 eV in the 1T-IrO₂ and from 2.7 eV in the rutile-IrO₂. This suggests that the thermal stability of rutile phase should be better than the 1T-phase, which agrees well with our experimental observations.

The Gibbs free energy profile of OER on 1T-IrO₂ is shown in Fig. 5d, e. Computational Pourbaix diagram indicates the reaction surface of 1T-IrO₂ mainly happens on the bare (0001) surface at U < 1.76 V/RHE (Supplementary Fig. 27) and the active site has been assigned to the Ir atoms. The binding of OH and the

subsequent *OH dissociation over Ir atoms are the potential-limiting steps. The formation of *OOH and $O_2$ molecules, however, are feasible. For the OER on a rutile-$IrO_2$ (110) surface, the corresponding limiting overpotential is 0.59 V, which agrees well with previous calculations (Fig. 5f, g)[46]. By comparing the free energy profiles of OER on the active sites discussed above, the superior activity of 1T-$IrO_2$ can be interpreted with the Sabatier's principle. On the surfaces of metal oxides, the binding of OH groups with the exposed metal ions are usually too strong. As a consequence, the subsequent formation of *O will cost high energies. On ideal OER catalysts, however, the binding of OH on the active site should exhibit an energy increase of 1.23 eV. From our calculations, it is clearly shown that the formations of *OH on the active sites of rutile-$IrO_2$ (110) surfaces are either exothermic or slightly endothermic. By contrast, this step is much more endothermic on the 1T-$IrO_2$. Therefore 1T-$IrO_2$ provides the active site, different from rutile-$IrO_2$, which leads to the enhanced catalytic performance. As shown in Supplementary Fig. 28, Ir atom on the surface of 1T-$IrO_2$ is buried under oxygen atoms, which leads to the appropriate adsorption of hydroxyl and easy desorption step. However, Ir atom on the surface of rutile-$IrO_2$ is not buried under oxygen atoms, which leads to the strong adsorption of hydroxyl and hard desorption step. The 1T-$IrO_2$ is thus more active than rutile-$IrO_2$ in OER.

## Discussion

In conclusion, a metallene oxide, 1T-$IrO_2$, has been successfully prepared by the mechano-thermal method. The optimized Ir active site in 1 T-phase and ultrathin 2D structure are the keys to the high-performance acidic OER catalysis. Computational calculations reveal that the catalytic origin of 1T-$IrO_2$ in OER can be attributed to the optimized free energy increase of binding hydroxyl groups on Ir atoms. The present research paves out a mechano-thermal method for synthesizing 2D materials and shows great promise for sustainable energy toward real applications and fundamental insight into physical understanding.

## Methods

**Chemicals.** Iridium trichloride ($IrCl_3$, 99.9%) was purchased from Alfa Aesar Co. Iridium(IV) oxide ($IrO_2$, 99%) was purchased from Aladdin Chemical Regent Co. Potassium hydroxide (KOH, 99%), sulphuric acid ($H_2SO_4$, ≥ 96.0%), and perchloric acid ($HClO_4$, 70.0–72.0%) were obtained by Sinopharm Chemical Reagent Co. Nafion solution (5 wt %) and mercury(II) nitrate hydrate ($Hg(NO_3)_2 \cdot H_2O$, 99.99%) were obtained from Sigma–Alddrich Co. Isopropanol was purchased from Sinopharm Chemical Reagent Co. Other reagents were of analytical reagent grade without further purification. Double-distilled water was used throughout the experiment. 20% Ir on Vulcan XC-72 was purchased from Premetek Co. The PEM device was purchased from Gaossunion (TIANJIN) Photoelectric Technology CO., LTD.. A commercial Nafion N117 membrane was purchased from Alfa Aesar.

**Synthesis of 1T-$IrO_2$.** The synthesis of 1T-$IrO_2$ was carried out in a home-made mechano-thermal reactor. The home-made mechano-thermal reactor is made by the high-purity corundum. Two hundred fifty milligrams of $IrCl_3$ and 10 g KOH were ground continuously with annealing from room temperature to 800 °C in a home-made mechano-thermal reactor. The heating rate is 10 °C min$^{-1}$. The rotation speed for the mechano-thermal reaction is 6 rpm. The reaction temperature was kept at 800 °C for 3 h, and then naturally cooled to room temperature. The as-prepared products were washed by double-distilled water for several times and dried by lyophilization to obtain 1T-$IrO_2$. Rutile-$IrO_2$ was synthesized by annealing 1T-$IrO_2$ at 900 °C for 2 h.

**Characterizations.** The XRD patterns of all samples were recorded by X-ray powder diffraction (XRD, Empyrean) by using the operation voltage of 40 kV and the current of 40 mA with Cu Kα radiation source. The TEM images of all samples were characterized by a FEI Tecnai F20 transmission electron microscope with an accelerating voltage of 200 kV and by an aberration-corrected JEOL ARM 200 F microscope operated at 200 kV. EDX was conducted by FEI Tecanai F20 transmission electron microscope. The SEM images were recorded by Zeiss G500 scanning electron microscope. The XPS results were collected by Kratos AXIS UltraDLD ultrahigh vacuum surface analysis system with Al Kα radiation (1486 eV). AFM images were captured by the atomic force microscopy (AFM,

Bruker Dimension Icon). Synchrotron X-ray absorption spectroscopy data were collected at beamline 7-BM at National Synchrotron Light source II at Brookhaven National Laboratory and the corresponding data are analyzed based on the ref. [39]. The BET specific surface areas were characterized by American Micromeritics ASAP-2020 porosimeter. The Zn content and Ir dissolution concentration were determined by aurora M90 ICP-MS. The produced $O_2$ was measured by the gas chromatograph (GC Agilent 7890B).

**Electrochemical measurements.** All electrochemical experiments were carried out with using the CHI660E electrochemical workstation (Shanghai Chenhua, China). A traditional three-electrode system was used. The glass-carbon electrode (GCE, area: 0.0707 cm$^2$ or 1 × 1 cm) was used as the working electrode, calomel electrode was used as the reference electrode, and carbon rod was used as the counter electrode. OER measurements were conducted in $O_2$-saturated 0.1 M $HClO_4$ or 0.5 M $H_2SO_4$ electrolyte. For the measurement with iR-correction, R was referred to the ohmic resistance arising from the electrolyte/contact resistance of the setup. The scan rate was 5 mV s$^{-1}$ when LSV was measured. The catalysts solutions were obtained by mixing 1.5 mg catalysts (1T-$IrO_2$, Rutile-$IrO_2$, C-$IrO_2$ or C-Ir/C) in the solution of 400 μL isopropanol and 10 μL 0.5 wt % Nafion solution with sonication to form the homogenous catalysts ink. 4 μL ink was dispersed on the GCE with drying naturally for testing. For the scale-up experiment, the catalysts inks were obtained by mixing 0.85 mg catalyst in the solution of 200 μL isopropanol and 10 μL 0.5 wt % Nafion solution with sonication to form homogenous catalysts ink. All the ink was dispersed on the 1 × 1 cm glass-carbon electrode with drying naturally for the scale-up testing. The XRD result of the 1T-$IrO_2$ after stability test was obtained by analyzing the electrocatalyst dropping on the carbon paper after stability test. The catalytic activity and durability were measured at least three times for each catalyst. Errors are reported as relative standard deviation (RSD).

For the PEM device, the anodic 1T-$IrO_2$ (C-$IrO_2$) and cathodic Pt/C (20%, commercial) were dropped on carbon paper with the active area of 1 × 1 cm, respectively. The catalyst ink was obtained by mixing 0.85 mg 1T-$IrO_2$ (C-$IrO_2$) in the solution of 200 μL isopropanol and 10 μL 0.5 wt % Nafion solution with sonication to form the homogenous catalyst ink. The electrolyte is $O_2$-saturated 0.5 M $H_2SO_4$ (3 mL min$^{-1}$) and the reaction temperature is 65 °C. N117 membrane was used for PEM device.

The ECSAs of 1T-$IrO_2$, Rutile-$IrO_2$ and C-$IrO_2$ were calculated by the Eqs. 1,2[47] (Supplementary Fig. 11a–d and Supplementary Table 5):

$$R_f \left( \text{Roughness factor} \right) = \frac{C_{dl}}{C_s} \tag{1}$$

$$\text{ECSA} = R_f \times A_{geo} \tag{2}$$

The electrochemical double-layer capacitance ($C_{dl}$) at non-Faradic potential range was obtained by measuring the capacitance of double layer at solid-liquid interface employing cyclic voltammetry (CV) with different scan rates (5, 10, 15, 20, 25, 30, and 35 mV s$^{-1}$) in a range from 1.0 and 1.1 V vs. RHE. The $C_{dl}$ was calculated by the Eq. 3:

$$C_{dl} = \frac{\Delta j/2}{\nu} \tag{3}$$

where $\Delta j$ is the difference between the anodic and cathodic currents ($j_a - j_c$) at 1.05 V vs. RHE and $\nu$ is the scan rate. Therefore, once the values of $A_{geo}$, $C_s$, and $C_{dl}$ are determined, the value of ECSA can be calculated according to Eqs. 1 and 2.

The $C_s$ values were calculated based on the method in the previous report[48], where 0.88 mF cm$^{-2}$ was calculated for 1T-$IrO_2$, 0.32 mF cm$^{-2}$ was calculated for Rutile-$IrO_2$, and 0.42 mF cm$^{-2}$ was calculated for C-$IrO_2$ (Supplementary Table 3 and Supplementary Note 1).

The ECSA of C-Ir/C was determined by Eqs. 4,5 (HUPD method)[49] (Supplementary Fig. 11e).

$$\text{ECSA} = \frac{Q}{C} \tag{4}$$

$$Q = \frac{S_{peak}}{\nu} \tag{5}$$

C is a value of the adsorbed capacity of hydrogen (210 μC cm$^{-2}$). $S_{peak}$ is the integral area of adsorbed hydrogen in the CV curve and $\nu$ is the scan rate of 5 mV s$^{-1}$.

The method of obtaining Hg underpotential deposition derived surface area is based on the ref. [50]. The catalyst ink for Hg underpotential deposition was prepared by mixing 3.5 mg catalyst with the solution of 7.8 mL double-distilled water, 2.2 mL isopropanol and 400 μL 0.5 wt % Nafion solution. Ten microliters of the ink was pipetted on the GCE for testing. The concentration of mercury nitrate was 1.0 mM.

According to Eq. 6, TOF is defined as the ratio between the number of total oxygen turnover per geometric area and the number of active sites per geometric area[51].

$$\text{TOF} = \frac{\#\text{total } O_2 \text{ turnover/cm}^2 \text{ per geometric area}}{\#\text{active sites/cm}^2 \text{ per geometric area}} \tag{6}$$

The total number of oxygen turnover is derived from according to Eq. 7.

$$\#_{O_2} = \left(j \frac{mA}{cm^2}\right) \times \left(\frac{1\,Cs^{-1}}{1000\,mA}\right) \times \left(\frac{1\,mol\,e^{-1}}{96485\,C}\right) \times \left(\frac{1\,mol\,O_2}{4\,mol\,e^{-}}\right) \times \left(\frac{6.023 \times 10^{23} O_2\,molecules}{1\,mol\,O_2}\right) \tag{7}$$

$$= 1.56 \times 10^{15} \frac{\frac{O_2}{s}}{cm^2} per \frac{mA}{cm^2}$$

Equation 8 determines the average active surface atoms on the terrace surface of ultrathin 1T-IrO$_2$.

$$\#active\,sites = \frac{0.5\,atom}{(3.11 \times 3.11)\,\text{Å}}^2 \times \sin 60^\circ = 5.97 \times 10^{14} \frac{atoms}{cm^2} \tag{8}$$

TOF of 1T-IrO$_2$ was calculated as follows (Eq. 9).

$$TOF_{UPD/BET/ECSA} = \frac{\left(1.56 \times 10^{15} \frac{O_2/s}{cm^2} per \frac{mA}{cm^2}\right) \times |j|}{(active\,sites) \times A_{UPD/BET/ECSA}} \tag{9}$$

The TOFs of Rutile-IrO$_2$, C-IrO$_2$, and C-Ir/C were calculated according to the same procedure.

**DFT calculations.** The spin polarized DFT calculations were carried out using the Vienna Ab-initio Simulation Package (VASP version 5.4)[52]. The exchange-correlation functional was described by the Perdew, Burke, and Ernzerhof (PBE) parameterization of the generalized gradient approximation (GGA)[53]. The electron-ion interactions were treated within the projector augmented-wave (PAW) approximation[54]. A cutoff energy of 400 eV was employed for the plane-wave basis set of all calculations. A conjugate-gradient algorithm was used to relax the atoms into their instantaneous ground state positions. The structural optimizations were not stopped until the ionic relaxation reaching the convergence criteria of $1 \times 10^{-5}$ eV and 0.01 eV Å$^{-1}$. The 1T-IrO$_2$ surface was modelled with a double layered $(4 \times 4)$ supercell. The rutile-IrO$_2$ (110) surface was mimicked by a 4-layer $(1 \times 2)$ periodic slabs, in which the top two layers are relaxed in three dimensions, and the bottom two layers are restricted. The first irreducible Brillouin zone was modelled with the Gamma-centered Monkhorst-Pack scheme, where a $4 \times 4 \times 1$ k-points and a $2 \times 2 \times 1$ k-points were adopted for 1T-IrO$_2$ (0001) and rutile-IrO$_2$ (110) surfaces, respectively. The vdW-D3 method was adopted to describe the van der Waals interactions[55]. A vacuum of 20 Å was employed to avoid the interactions between periodic images in the direction normal to the surface. The effect of aqueous solution was considered with the implicit solvation model, in which the relative permittivity was set to 80.00[56].

The change of the Gibbs free energy (ΔG) for each elementary step at the zero potential can be written as:

$$\triangle G(0) = \triangle E + \triangle E_{ZPE} - T\triangle S + \triangle G_{pH} \tag{10}$$

where E is the energy that can be directly obtained from the DFT calculations. $E_{ZPE}$ is the zero-point energy that can be calculated by $E_{ZPE} = 1/2\Sigma h\nu$, in which $\nu$ is the vibrational frequency of a normal mode and $h$ is the Planck constant. S is the entropy that can be calculated as[57]:

$$S(T) = k_B \sum_i \left( \frac{h\nu_i}{k_B T \left( \exp\left(\frac{h\nu_i}{k_B T}\right) - 1 \right)} - \ln\left(1 - \exp\left(\frac{-h\nu_i}{k_B T}\right)\right) \right) \tag{11}$$

$k_B$ and $\nu_i$ are the Boltzmann constant and vibrational frequency, respectively. $\Delta G_{pH}$ is the free energy contributions related to the H$^+$ concentration, which can be calculated by: $\Delta G_{pH} = 2.303 k_B TpH$[58,59]. In this work, the pH = 0.00 and T = 298.15 K were selected to represent the acidic media and room temperature. According to the computational hydrogen electrode (CHE) model proposed by Nørskov et al., the Gibbs free energy of the proton-electron pairs (H$^+$ + e$^-$) is equivalent to one-half of the Gibbs free energy of a hydrogen molecule[57]. ΔG(U) with an applied potential U can be defined as:

$$\triangle G(U) = \triangle G(0) - neU \tag{12}$$

where the $-neU$ is the change of the chemical potential of the proton and electron pair when the external potential U is applied. The overpotential ($\eta$) of the whole OER process can be described as:

$$\eta = U_{limiting} - U_{equilibrium} \tag{13}$$

where $U_{equilibrium}$ is 1.23 V and $U_{limiting}$ is the limiting potential of the OER. The limiting potential (onset potential) can be obtained by the following equation: $U_{limiting} = \Delta G_{max}/e$, where $\Delta G_{max}$ is the change of Gibbs free energy in the potential-limiting step (PLS). Considering the effect of the electrochemical electrode potential and the solvation, we also used JDFTx software[60] to calculate the $U_{limiting}$ at the constant electrode potential of 1.23 V along with the CANDLE implicit solvation model[61] and found that both of the constant-charge and constant-potential simulation results have the same trend. The quantitative effects of electrode potential and functionals on the $U_{limiting}$ are given in Supplementary Tables 13, 14.

## Data availability
The experiment data that support the findings of this study are available from the corresponding author upon reasonable request. Source data are provided with this paper.

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

## Acknowledgements

This work was financially supported by Development Program of China (2017YFA0204800), National MCF Energy R&D Program (2018YFE0306105), the National Natural Science Foundation of China (22025108, 21771134, 21905188), China Postdoctoral Science Foundation Grant (2019M651937), Jiangsu Key Laboratory for Carbon-Based Functional Materials & Devices, Soochow University (No. KJS2019), the Collaborative Innovation Center of Suzhou Nano Science & Technology, the Priority Academic Program Development of Jiangsu Higher Education Institutions (PAPD), the 111 Project and Joint International Research Laboratory of Carbon-Based Functional Materials and Devices. This research used 7-BM beamline of the National Synchrotron Light Source II and resources of the Center for Functional Nanomaterials, which are U.S. Department of Energy (DOE) Office of Science User Facility operated for the DOE Office of Science by Brookhaven National Laboratory under Contract No. DE-SC0012704. We also acknowledge the use of facilities within the Eyring Materials Center at Arizona State University supported in part by NNCI-ECCS-1542160.

## Author contributions

Q.S., X.H., and M.S. conceived and supervised the research. M.S. designed and fabricated the mechano-thermal reactor. Q.D., Q.S., X.H., and M.S. designed the experiments. Q.S., X.H., M.S., Q.D., Z.F., S.G., W.Z., and F.L. performed most of the experiments and data analysis. H.L., Y.J., F.Z., N.K., and Y.L. performed and analyzed the DFT simulations. S.Y. and L.M. characterized the material structure combining TEM/STEM, DFT and synchrotron X-ray absorption and proposed the structure model. Q.D., Q.S., X.H., and M.S. wrote the paper. All authors discussed the results and commented on the manuscript. Q.D., H.L., Z.F., and L.M. equally contribute to this work.

## Competing interests

The authors declare no competing interests.
