## [Peer Review File · Nature Communications]

Iridium Metallene Oxide for Acidic Oxygen Evolution CatalysisREVIEWER COMMENTS

Reviewer #1 (Remarks to the Author):

The submitted manuscript describes the synthesis of a 1T-phase iridium oxide material, which shows remarkable activity towards the OER in acid. This is a well-written, thorough report, from synthesis and structural characterization of a novel material, to its electrochemical characterization; finally, DFT studies provide insight into the high activity of this material. Several control experiments were thoroughly carried out, and the results well explained.

Overall, I recommend publication of this manuscript pending a few minor edits suggested below:

1) While the manuscript is mostly well-written, it would benefit at a second pass to improve grammar and syntax in a few places (this was evident in the methods section than the main body of the work).

2) The reaction conditions are not a standard method most researchers would be familiar with. It would improve the manuscript if the authors could provide more insight into how - and why - the highly alkaline environment and mechanical energy help in forming this new phase. In particular, I did not find that Fig. S1 and Scheme 1 clearly corresponded to one-another. The geometry seems different, which is not helpful in helping other researchers duplicate this work. While a nice feature of this work is that this new synthetic method is used, the same can also be seen as a limitation in the ease to reproduce and adapt this work by other groups. As such, the more details to enable reproducibility in those would be useful.

3) Table 8 formatting should be fixed for better readability.

Reviewer #2 (Remarks to the Author):

The authors report the synthesis of an IrO₂ catalyst with few-layer ultrathin sheet morphology and compare it with rutile-IrO₂ and Ir-based commercial catalysts for oxygen evolution reaction in acid. The catalyst is well characterized in details and tested both in a three-electrode configuration as well as PEM electrolyser. The enhanced performance is supported by density functional theory calculations. The synthesis of this active and stable 2D-OER catalyst is interesting for both the community working on improving PEM electrolyser for generating hydrogen and the understanding of the properties of 2D materials in relation to catalysis and electrocatalysis.

Some parts of the text and some details need to be clarified. Thus, I recommend to publish after minor revision.

Main comments:

Introduction: In my opinion the authors' strategy (i.e. 2D material) should be compared with other main strategies in the field that aim to improve the Ir-based catalysts for OER, to place the authors' work in the state-of-the-art (SoA) literature. The authors quantitatively compare the performance of their catalysts with other SoA catalysts in SI, which is appreciated, but I think there should be also a qualitative discussion in the introduction.

Page 5, line 4: "(AFM) analyses also confirm the ultrathin 2D structure of 1T-IrO₂". The morphology and thickness of the catalyst sheets are very important for this work, where the 2D nature of the catalyst is highlighted. Therefore, I think the authors should add in the main text some more indication related to this point, not only refer to SI. For example, a simple addition would be "... structure of 1T-IrO₂, with thicknesses of 3-5 nm (Supplementary Fig.3)". This will not increase the length too much.

Page 5, line 19: "from monolayer region". It has not been introduced before that there are multiple regions, or what is the thickness of these catalysts. I suggest, for example, to add at line 7 "...cross-sectional view showing monolayer and bilayer regions, and linear..." or other expressions.

Page 6, line 10: "We adopted the H and T phase... converged very well.". Where are these results?

Page 16, line 6: "The CS values were calculated based on the method in previous report...". The calculation of Cs is not clear to me, even with the values indicated in the Supporting Table 2 and 3. Is the calculation of "Catalyst Area" in Supp. Table 2 involving the use of constants? I couldn't follow the calculation looking into the mentioned reference by Savinell et al.. And once the Catalyst Area is derived, how the authors obtain the corresponding capacitance, which I guess is what is defined here as CS and indicated in Supp. Table 3? I suggest to improve the description of this procedure.

Page 17, line 6, Equation 8: I understand that the authors assume only the terrace surface to calculate the number of active sites, which is probably reasonable considering the ultrathin morphology. Nonetheless, this should be indicated.

Page 18, line 4 and DFT part in the main text: The (110) surface is specified for the rutile- IrO_2 , but no surface is specified for the 1T- IrO_2 . It is important to specify the surface also in that case, because the edges and the terraces most likely will provide active sites with different overpotentials and for future comparison of results.

Some other minor comments:

Page 2, line 19: "more importantly, increasing demands... industrial scale". This sentence is not super clear to me. Changing the word "demands" with some other expressions might make it clearer, in my opinion.

Page 3, line 11: "As the surface energy increases ... shows the dimension-dependent behavior". The authors use the expression "the dimension-dependent behavior", which should refer to a specific behavior that they mentioned before or that it is universally known. But I have not found this description before. I suggest to briefly specify this behavior here.

Page 6, line 14: "Fig.2f". Figure 2f, 2g and 2h are described in the text before Fig 2e and 2a-d. Shouldn't the figure panels then be rearranged?

Page 9, line 19: "in the scale-up experiment". If I am right, the scale up-experiment has not been introduced before. It is described in the method section. So, the authors should add either a reference here to the Method section or shortly mention the difference, i.e. "in the scale-up experiment performed with larger 1x1 cm electrodes".

Supp. Table 3: "Css" probably should be "Cs".

Page 10, line 23: "that the uniform". I think that "that" should be removed since there is not a verb after. As well as at line 11, page 11: "confirms that the".

Page 12, line 1: Probably, "they are likely" to "become".

Page 12, line 14: "respectively" indicates more values, but I see only 0.59V, I think that a value is missing here.

Reviewer #3 (Remarks to the Author):

The authors present a new Iridium based anode catalyst material for the electrolytic water splitting reaction. The material has been synthesized using a combination of milling, high temperature and harsh alkaline conditions and is characterized as 2D material. The authors describe the material to exhibit long term stability and an overpotential of 0.2 V for the OER, which, if this is the

case, would be truly spectacular.

The material is analyzed using a broad spectrum of methods including XRD, STEM-ADF, XANES, EXAFS, DFT optimizations, and XPS. From the results, the authors derive a structure referred to as 1T-IrO₂ which is supposed to exhibit a bulk structure very different from the typical Rutile phase. They attribute the superior electrocatalytic properties to this structure and support their claims with a computational investigation of the free energies of adsorption of OH, O and OOH following Norskov's scheme and "volcano plot" arguments.

As water splitting is one of the key technologies for storing renewable energies and industry all over the globe is working on large-scale technical solutions, I would expect this work to have a significant impact and I would also consider it of interest for a broader audience.

Before stating my comments about the work I must clarify that I am a computational chemist and can only thoroughly judge on the theory part of the work. I would consider the findings reported in the paper worth publishing in Nature Communications if reviewers with experimental and electrochemical background agree on the quality of the work. The manuscript in its current form would require a major revision, though.

1) The language needs substantial improvement - there are several incomprehensible statements and many awkward formulations which make reading the manuscript difficult and several paragraphs unclear. I strongly recommend revision, if possible with the help of a native speaker.

2) While most of the arguments of the authors concerning the structure of the raw catalyst material are supported by a large variety of investigations, there is a lack of surface characterization. Most techniques applied focus on the bulk structure, but the essential interface (defects, termination etc.) has not been studied in too much detail. As this is the fundamental input for the computations, some confidence in what surface termination the material has in aqueous solution (or at least in ambient conditions) would be helpful.

3) The material is synthesized in alkaline conditions, but used as catalyst in acidic medium and at high electrochemical potentials. As the stress under working conditions is high, and even noble metal catalysts are known to undergo structural changes, the post-analysis the authors present is welcome, but also not really sufficient. Besides this, operando characterization is limited to an ICP-MS investigation for which only a single table is given. Here, much more details about Ir dissolution, O₂ evolution and potential cycling experiments should be given.

4) Commercial IrO₂ catalysts typically require pre-conditioning before using as catalysts for the OER, because they often contain Ir in various oxidation states, sometimes mixed with support materials like TiO₂. In many cases only upon potential cycling, an active and stable material is obtained. This is missing from the comparative studies the authors present and some more results along these lines would be required in order to ensure that the new material is a) superior to the other catalyst materials and b) can withstand start-stop cycles in electrolysis.

5) The stability test (p. 10) is only carried out with 1T-IrO₂ and C-IrO₂ but not with the other reference materials. This should be changed and the other materials (Rutile, C-Ir/C) should be included and compared.

6) The computational investigation on the structure of the 1T-IrO₂ (page 6, second paragraph) is sound but the authors should express some words of caution - just because a structure is a local minimum, it does not mean it is stable in solution or at higher potential. Furthermore, structures have only been optimized for the bulk, as far as I understand. As for catalysis the interface is essential, DFT calculations should also focus on surface stability for different surface terminations (ab-initio thermodynamics / computational Pourbaix diagrams, see work by Scheffler, Gross, Todorova etc.).

7) The computational study of the "reactivity" of the material (Fig. 4) apply conventional DFT methods and an a-posteriori approach for the inclusion of the electrochemical potential. While the authors did choose to include implicit solvent and the results are consistent and allow for some degree of comparison, this approach is not really state of the art anymore. Simply computing adsorption energies of molecular and atomic species is a rough indicator, but the electronic structure of the material is undefined in terms of the electrochemical potential. Modern investigations using electronic structure theory for OER and ORR should explicitly include the effect of the electrochemical potential in the simulations (Work by Goddard, Neugebauer, Bhattacharyya etc.) and give a more accurate picture of structures, energetics and reactivity. In order to live up to the high standards of Nature Communications, the authors should adapt a more accurate description of the underlying physics in their models. This way, the simulation would also yield more details on how the material behaves and how reactivity is influenced by the potential. Currently, all values are energies of neutral systems including implicit solvation, which are scaled a-posteriori for a rough estimate of the potential influence.

8) While computational details are given in the manuscript and one figure on the computed structures is given in the SI, no further details are provided. All simulated structures should be given in the SI for reproducibility. Furthermore, the authors have used one Functional for all calculations - here it would be desirable to also apply other functionals and report differences so that the reader can assess deviations.

Response to Reviewers

Dear Reviewers,

Thank you for your precious time to constructive comments on our manuscript titled “Iridium Metallene Oxide for Acidic Oxygen Evolution Catalysis” (Manuscript ID: NCOMMS-21-26540-T) for Nature Communications. We sincerely appreciate your comments and suggestions on our work, which are highly important for the further improvements to our manuscript. According to all the comments, we have made a detailed response and substantial revisions in our revised manuscript.

Reviewer #1

The submitted manuscript describes the synthesis of a 1T-phase iridium oxide material, which shows remarkable activity towards the OER in acid. This is a well-written, thorough report, from synthesis and structural characterization of a novel material, to its electrochemical characterization; finally, DFT studies provide insight into the high activity of this material. Several control experiments were thoroughly carried out, and the results well explained.

Overall, I recommend publication of this manuscript pending a few minor edits suggested below:

[Author’s Response]: We would like to thank you for the positive reports and recommendation of the publication in the Nature Communications. Your comments lead to further improve the quality of our work. According to your comments, we have modified our manuscript discussion and corresponding responses.

1. While the manuscript is mostly well-written, it would benefit at a second pass to improve grammar and syntax in a few places (this was evident in the methods section than the main body of the work).

[Author’s Response]: Thank you for your valuable opinion. We have modified the manuscript, especially the Method Sections in the new revision.

2. The reaction conditions are not a standard method most researchers would be familiar with. It would improve the manuscript if the authors could provide more insight into how - and why - the highly alkaline environment and mechanical energy help in forming this new phase. In particular, I did not find that Fig. S1 and Scheme 1 clearly corresponded to one-another. The geometry seems different, which is not helpful in helping other researchers duplicate this work. While a nice feature of this work is that this new synthetic method is used, the same can also be seem as a limitation ion the ease to reproduce and adapt this work by other groups. As such, the more details to enable reproducible method would be useful.

[Author’s Response]: Thank you for your valuable opinion.

1T-IrO₂ is a metastable phase of IrO₂. The growth of metastable phase materials requires higher formation energy than those of thermodynamically stable phase materials. Therefore, external energy should be supplied to grow the metastable materials, such as mechanical energy, making it difficult to obtain the metastable materials in the standard synthetic conditions. Here, mechanochemical synthesis provides high energy input during the synthetic progress. As shown in **Supplementary Figure 4a**, only the amorphous product was obtained without applying mechanical energy.

In addition, according to Liao’s report (Liao, J. H. et al. Synthesis and crystal growth of two novel layered structures, NaKLaNbO₅ and Na₂K₂Gd₄Nb₂O₁₃, in molten hydroxide salts. Cryst. Growth. Des. 2, 83-85 (2002)), the molten alkali metal hydroxides have been described as a Lux acid-base solvent system, which is suitable for the synthesis of metal oxides. As suggested, we also carried out

new synthesis reactions by changing the alkaline precursors. When using different alkali ions (LiOH or NaOH), only the nanoparticles can be obtained (**Fig. R1**), suggesting the important role of KOH in forming the 1T phase 2D materials.

Second, as suggested, we have revised the **Scheme 1** for better understanding.

Figure R1. SEM images of samples using different alkalis. **a,b**, Products by using LiOH and IrCl₃ as raw materials. **c,d**, Products by using NaOH and IrCl₃ as raw materials. The reaction temperature is 800 °C and the reaction time is 3 h.

Scheme 1. Schematic representation of the mechano-thermal reactor for preparing 1T-IrO₂, where the mechanical and thermal operations are controlled simultaneously. The blue and yellow balls indicate IrCl₃ and KOH, respectively.

3. Table 8 formatting should be fixed for better readability.

[Author's Response]: Thank you for your valuable opinion. **Supporting Table 8** has been fixed in the new revision.

Reviewer #2

The authors report the synthesis of an IrO₂ catalyst with few-layer ultrathin sheet morphology and compare it with rutile-IrO₂ and Ir-based commercial catalysts for oxygen evolution reaction in acid. The catalyst is well characterized in details and tested both in a three-electrode configuration as well as PEM electrolyser. The enhanced performance is supported by density functional theory calculations. The synthesis of this active and stable 2D-OER catalyst is interesting for both the community working on improving PEM electrolyser for generating hydrogen and the understanding of the properties of 2D materials in relation to catalysis and electrocatalysis.

Some parts of the text and some details need to be clarified. Thus, I recommend to publish after minor revision.

[Author's Response]: Thank you for your positive comments. Your comments lead to further improve the quality of our work. According to your comments, we have modified our manuscript discussion and corresponding responses.

Main comments:

1. Introduction: In my opinion the authors' strategy (i.e. 2D material) should be compared with other main strategies in the field that aim to improve the Ir-based catalysts for OER, to place the authors' work in the state-of-the-art (SoA) literature. The authors quantitatively compare the performance of their catalysts with other SoA catalysts in SI, which is appreciated, but I think there should be also a qualitatively discussion in the introduction.

[Author's Response]: Thank you for your opinion. As suggested, we have added the discussion about the other main strategies that used to improve the Ir-based catalysts for OER. The related discussion has been added in the introduction part.

[Added Content]: [Page 3, Line 1] "Several strategies have been reported to develop complex iridium-based oxides, such as: surface restructuration strategy¹⁷, amorphous strategy¹⁹ and doping strategy¹⁸, yet still leaving a large room for improvement."

2. Page 5, line 4: "(AFM) analyses also confirm the ultrathin 2D structure of 1T-IrO₂". The morphology and thickness of the catalyst sheets are very important for this work, where the 2D nature of the catalyst is highlighted. Therefore, I think the authors should add in the main text some more indication related to this point, not only refer to SI. For example, a simple addition would be "... structure of 1T-IrO₂, with thicknesses of 3-5 nm (Supplementary Fig.3)". This will not increase the length too much.

[Author's Response]: Thank you. The manuscript have modified according to your valuable suggestion (**Page 5, Line 6**).

3. Page 5, line 19: "from monolayer region". It has not been introduced before that there are multiple regions, or what is the thickness of these catalysts. I suggest, for example, to add at line 7 "...cross-sectional view showing monolayer and bilayer regions, and linear..." or other expressions.

[Author's Response]: Many thanks for the suggestion. We have updated the paragraph as suggested (**Page 5, Line 9**).

4. Page 6, line 10: "We adopted the H and T phase... converged very well.". Where are these results?

[Author's Response]: Many thanks for the comments. We have used the structural models of

1T-MoS₂ and 1H MoS₂ provided in the reference paper (Yang, S.-Z. et al. Rhenium-doped and stabilized MoS₂ atomic layers with basal-plane catalytic activity. *Adv. Mater.* 30, 1803477 (2018).) and replaced Mo and S atoms with Ir and O atoms. We also confined the unit cell size to the experimental value ($a = 3.11 \text{ \AA}$ and $c = 6.91 \text{ \AA}$) from XRD and electron diffraction measurement (**Fig. R2**). We then applied the full structural relaxation in VASP which allows all the atoms to move while fixing the unit cell size. By setting a convergence value of 0.01 eV, after relaxation, both initial structures will converge to the 1T structure.

Figure R2. The crystal structures of (a) 1H-IrO₂ and (b) 1T-IrO₂. The yellow ball is Ir and the red ball is oxygen.

5. Page 16, line 6: “The CS values were calculated based on the method in previous report...”. The calculation of C_s is not clear to me, even with the values indicated in the Supporting Table 2 and 3. Is the calculation of “Catalyst Area” in Supp. Table 2 involving the use of constants? I couldn’t follow the calculation looking into the mentioned reference by Savinell et al.. And once the Catalyst Area is derived, how the authors obtain the corresponding capacitance, which I guess is what is defined here as CS and indicated in Supp. Table 3? I suggest to improve the description of this procedure.

[Author’s Response]: Thank you for your valuable opinion.

Different catalysts have different values of C_s. Here, C_s values were calculated as followings: (Savinell, R. F. et al. Electrochemically active surface-area-voltammetric charge correlations for ruthenium and iridium dioxide electrodes. *J. Electrochem. Soc.* 137, 489-494 (1990))

$$C_s = \text{voltammetric charge} \div (\Delta V \times \text{Area}_{\text{catalyst}}),$$

1T-IrO₂ was first dispersed on Ti electrode. The voltammetric curves of this working electrode were then collected in the voltage range (ΔV) from 0.05 - 1 V vs. SCE to obtain the voltammetric charges. The area of active site of catalyst ($\text{Area}_{\text{catalyst}}$) was then determined by the adsorption of Zn²⁺ ions (based on 17 Å² per Zn²⁺) onto the electrode surface (Kozawa, A. Ion-exchange adsorption of zinc and copper ions on silica. *J. Inorg. Nucl. Chem.* 21, 315-324 (1961)). 20 mg samples were dispersed in the 5 mL solution of 0.5 M NH₄Cl and 0.00122 M ZnO, and stayed for 16 h. The absorbed amount of Zn²⁺ ions was determined by inductively coupled plasma source mass spectrometer (ICP-MS). The description of this procedure is modified in the new revision in the **Supplementary Note 1**.

6. Page 17, line 6, Equation 8: I understand that the authors assume only the terrace surface to calculate the number of active sites, which is probably reasonable considering the ultrathin morphology. Nonetheless, this should be indicated.

[Author’s Response]: The manuscript have modified according to your valuable suggestion (**Page 18, Line 5**). Thank you!

7. Page 18, line 4 and DFT part in the main text: The (110) surface is specified for the rutile-IrO₂, but no surface is specified for the 1T-IrO₂. It is important to specify the surface also in that case, because the edges and the terraces most likely will provide active sites with different overpotentials and for future comparison of results.

[Author's Response]: In this work, the (0001) surface of 1T-IrO₂ and the (110) surface of rutile IrO₂ are adopted to simulate the four-electron OER processes. Meanwhile, we also analyzed the Pourbaix diagram of 1T-IrO₂ (0001) and found that the bare 1T-IrO₂ (0001) surface was the actual reaction surface at U=1.23 V/RHE condition. The reason why we chose the rutile-IrO₂ (110) and 1T-IrO₂ (0001) surfaces is that these two surfaces have the lowest formation energies, which are the main exposed surface for the OER although other high-index surfaces may have better OER activities.

[Added Content]: [Page 19, Line 1] “The first irreducible Brillouin zone was modelled with the Gamma-centered Monkhorst-Pack scheme, where a $4 \times 4 \times 1$ k-points and a $2 \times 2 \times 1$ k-points were adopted for 1T-IrO₂ (0001) and rutile-IrO₂ (110) surfaces, respectively.”

8. Page 2, line 19: “more importantly, increasing demands... industrial scale”. This sentence is not super clear to me. Changing the word “demands” with some other expressions might make it clearer, in my opinion.

[Author's Response]: The manuscript have modified according to your valuable suggestion. Thank you!

[Added Content]: [Page 2, Line 19] “More importantly, it is required for developing OER catalysts in acidic pH regime for their direct impact on making the proton exchange membrane (PEM) viable on an industrial scale.”

9. Page 3, line 11: “As the surface energy increases ... shows the dimension-dependent behavior”. The authors use the expression “the dimension-dependent behavior”, which should refer to a specific behavior that they mentioned before or that it is universally known. But I have not found this description before. I suggest to briefly specify this behavior here.

[Author's Response]: The manuscript have modified according to your valuable suggestion. Thank you!

[Added Content]: [Page 3, Line 12] “As the surface energy increases with thinning the crystal thickness^{31,32}, the stability of phase increases with decreasing the layer thickness.

31. Chen, C.-C., Herhold, A. B., Johnson, C. S. & Alivisatos, A. P. Size dependence of structural metastability in semiconductor nanocrystals. *Science* **276**, 398-401 (1997).

32. Rodionov, A., Kalendarev, R. & Tchikvaidze, G. A new phase in solid state arsenic. *Nature* **281**, 60 (1979).”

10. Page 6, line 14: “Fig.2f”. Figure 2f, 2g and 2h are described in the text before Fig 2e and 2a-d. Shouldn't the figure panels then be rearranged?

[Author's Response]: Thank you. We have updated the **Fig. 2** for better understanding.

Figure 2. Structure representations of 1T-IrO₂. **a**, Ir-LIII edge XANES of 1T-IrO₂, rutile-IrO₂, and Ir foil. **b**, Ir-LIII edge XANES of T-phase IrO₂ and H-phase IrO₂ are simulated using Feff9 code and compared with experimental result of 1T-IrO₂. **c**, k²-Normalized Ir-LIII edge EXAFS of 1T-IrO₂, rutile-IrO₂ and Ir foil. **d**, k²-Normalized Ir-LIII edge EXAFS of 1T-IrO₂, where the Ir-O₆ and Ir-Ir₆ shells are highlighted. **e**, Atomic structure of layered 1T-IrO₂. **f-h**, Schematic of the unit cell of 1T-IrO₂ with the lattice constants of $a = b = 3.11 \text{ \AA}$ and $c = 6.91 \text{ \AA}$. The re-fined structure models are obtained based on the XRD and the aberration-corrected HAADF-STEM analysis. Ir and O are represented by blue and red spheres, respectively.

11. Page 9, line 19: “in the scale-up experiment”. If I am right, the scale up-experiment has not been introduced before. It is described in the method section. So, the authors should add either a reference here to the Method section or shortly mention the difference, i.e. “in the scale-up experiment performed with larger 1x1 cm electrodes”.

[Author’s Response]: Thank you. The manuscript have modified according to your valuable suggestion (Page 9, Line 22; Page 16, Line 4).

12. Supp. Table 3: “Css” probably should be “Cs”.

[Author’s Response]: Thank you. The manuscript have modified according to your valuable suggestion.

13. Page 10, line 23: “that the uniform”. I think that “that” should be removed since there is not a verb after. As well as at line 11, page 11: “confirms that the”.

[Author’s Response]: Thank you. The manuscript have modified according to your valuable suggestion (Page 11, Line 7; Page 11, Line 21).

14. Page 12, line 1: Probably, “they are likely” to “become”.

[Author’s Response]: Thank you. The manuscript have modified according to your valuable suggestion.

15. Page 12, line 14: “respectively” indicates more values, but I see only 0.59V, I think that a value is missing here.

[Author’s Response]: Thank you. The manuscript have modified according to your valuable suggestion (**Page 13, Line 6**).

Reviewer #3

The authors present a new Iridium based anode catalyst material for the electrolytic water splitting reaction. The material has been synthesized using a combination of milling, high temperature and harsh alkaline conditions and is characterized as 2D material. The authors describe the material to exhibit long term stability and an overpotential of 0.2 V for the OER, which, if this is the case, would be truly spectacular.

The material is analyzed using a broad spectrum of methods including XRD, STEM-ADF, XANES, EXAFS, DFT optimizations, and XPS. From the results, the authors derive a structure referred to as 1T-IrO₂ which is supposed to exhibit a bulk structure very different from the typical Rutile phase. They attribute the superior electrocatalytic properties to this structure and support their claims with a computational investigation of the free energies of adsorption of OH, O and OOH following Norskov's scheme and "volcano plot" arguments.

As water splitting is one of the key technologies for storing renewable energies and industry all over the globe is working on large-scale technical solutions, I would expect this work to have a significant impact and I would also consider it of interest for a broader audience.

Before stating my comments about the work I must clarify that I am a computational chemist and can only thoroughly judge on the theory part of the work. I would consider the findings reported in the paper worth publishing in Nature Communications if reviewers with experimental and electrochemical background agree on the quality of the work. The manuscript in its current form would require a major revision, though.

[Author's Response]: We would like to thank you for your valuable comments and suggestions, which lead to further improve the quality of our work. Following your comments and suggestions, we have modified our manuscript. All the modifications have been highlighted in the revised manuscript. We response all of your points one-by-one below.

1. The language needs substantial improvement - there are several incomprehensible statements and many awkward formulations which make reading the manuscript difficult and several paragraphs unclear. I strongly recommend revision, if possible with the help of a native speaker.

[Author's Response]: Thank you for your valuable suggestion. The manuscript has been modified by a native speaker in the new revision.

2. While most of the arguments of the authors concerning the structure of the raw catalyst material are supported by a large variety of investigations, there is a lack of surface characterization. Most techniques applied focus on the bulk structure, but the essential interface (defects, termination etc.) has not been studied in too much detail. As this is the fundamental input for the computations, some confidence in what surface termination the material has in aqueous solution (or at least in ambient conditions) would be helpful.

[Author's Response]: Thank you. The fundamental surface characterization is important. The XRD diffraction peaks of 1T-IrO₂ are strong (**Fig. 1d**). Surface characterization was also conducted: The SAED pattern clearly show that 1T-IrO₂ has high crystallinity (insert in **Fig. 1c**) and HAADF-STEM images (**Fig. 1e and 1f**) exhibit the regular atomic arrangement.

As to surface termination, XPS and EDX only show the existence of Ir and O, with no Cl and Ksignal. This EDX spectrum was collected on a transmission electron microscope, also a means of surface characterization. Since the synthesis process is carried out in an oxidizing environment:

exposing to air, strong alkaline starting material and high reaction temperature (10 g KOH and 800 °C), the surface of 1T-IrO₂ is terminated by O atoms. H element might emit in the form of water vapor at such a high temperature of 800 °C. All these results fit well with the calculation structure.

3. The material is synthesized in alkaline conditions, but used as catalyst in acidic medium and at high electrochemical potentials. As the stress under working conditions is high, and even noble metal catalysts are known to undergo structural changes, the post-analysis the authors present is welcome, but also not really sufficient. Besides this, operando characterization is limited to an ICP-MS investigation for which only a single table is given. Here, much more details about Ir dissolution, O₂ evolution and potential cycling experiments should be given.

[Author's Response]: Thank you. As suggested, SAED image was added, which shows that 1T-IrO₂ remain its high crystalline nature (**Supplementary Fig. 26a1**). X-ray absorption near-edge structure (XANES) of 1T-IrO₂ after the stability test was also carried out. As shown in **Supplementary Fig. 26g**, the valence state of 1T-IrO₂ does not change much compared to that of the reference rutile phase-IrO₂, suggesting its high stability.

In addition, we also test the Ir dissolution, O₂ evolution and potential cycling experiments to reveal stability property of 1T-IrO₂.

The calculation of s-number is according to the equation:

$$s - \text{number} = n_{O_2}(\text{OER}) \div n_{Ir}(\text{dissolved}),$$

$$n_{O_2}(\text{OER}) = (I \times S_{\text{electrode}} \times t \times \text{Faradic Efficiency}) \div (4 \times 96500),$$

where $S_{\text{electrode}}$ is 1 cm², the Faradaic efficiency is 100% according to the previous work (**Supplementary Figs. 23,24**),

$$n_{Ir}(\text{dissolved}) = V(50 \text{ mL}) \times \rho(1 \text{ g mL}^{-1}) \times C(\text{ppb}) \div M(192 \text{ g mol}^{-1}),$$

Form the data in **Supplementary Table 10**, all the s-numbers at different working potentials are greater than 1×10⁶. According to the reference, the s-number of commercial IrO_x catalyst reaches 5×10⁵ in acidic media (Speck, F. D. et al. Mechanisms of manganese oxide electrocatalysts degradation during oxygen reduction and oxygen evolution reactions. *J. Phys. Chem. C* 123, 25267-25277 (2019)). Such large values of s-number of 1T-IrO₂ confirm its excellent stability.

Supplementary Figure 26. **a**, The TEM image of 1T-IrO₂ after stability test in PEM device with the current density of 250 mA cm_{geo}⁻², clearly showing its sheet morphology. **a1**, The SAED pattern of 1T-IrO₂ in Supplementary Fig. 26a, showing the six-fold rotational symmetry. The scale bar in (**a**) is

100 nm. **g**, X-ray absorption near-edge spectroscopy spectra of 1T-IrO₂ after stability test and Reference rutile-IrO₂.

Supplementary Figure 18. The stability tests of 1T-IrO₂ under different potentials (vs. RHE).

Supplementary Table 10. ICP-MS results of 1T-IrO₂ under different potentials for 500 s.

Potential (V vs. RHE)	1.5	1.6	1.7	1.8	1.9	2.0
Time (s)	500	500	500	500	500	500
Concentration (ppb)	0.215	0.234	0.209	0.223	0.233	0.222
n _{O₂} (μmol)	72.8	81.7	110.7	131.2	147.2	153.5
s-number	1300000	1340000	2030000	2260000	2430000	2650000

[Added Content]: [Page 11, Line 19] “SAED image shows that 1T-IrO₂ remain its high crystalline nature (Supplementary Fig. 26a1).”

[Page 12, Line 1] “X-ray absorption near-edge structure (XANES) of 1T-IrO₂ after the stability test was carried out. As shown in Supplementary Fig. 26g, the valence state of 1T-IrO₂ does not change much compared to that of the reference rutile phase-IrO₂, suggesting its high stability.”

[Page 10, Line 22] “We also calculated the s-number to reveal stability property of 1T-IrO₂ (Supplementary Fig. 18). From the data in Supplementary Table 10, all the s-numbers at different working potentials are greater than 1×10^6 , larger than the s-number of commercial IrO_x catalyst (5×10^5) in acidic media⁴⁵.

45. Speck, F. D., Santori, P. G. Jaouen, F. & Cherevko, S. Mechanisms of manganese oxide electrocatalysts degradation during oxygen reduction and oxygen evolution reactions. *J. Phys. Chem. C* **123**, 25267-25277 (2019).”

4. Commercial IrO₂ catalysts typically require pre-conditioning before using as catalysts for the OER, because they often contain Ir in various oxidation states, sometimes mixed with support materials like TiO₂. In many cases only upon potential cycling, an active and stable material is obtained. This is missing from the comparative studies the authors present and some more results along these lines would be required in order to ensure that the new material is a) superior to the other catalyst materials and b) can withstand start-stop cycles in electrolysis.

[Author’s Response]: Thank you for your important suggestion. As suggested, new references are

added in **Supplementary Table 9** to prove the high stability of 1T-IrO₂ (**References S21,22,23**). In addition, the start-stop cycles test of PEM device was performed at 65 °C. The cycling was performed by repeatedly setting at 250 mA cm_{geo}⁻² for 300 s and then off for 300 s (open circuit). As shown in **Supplementary Fig. 25**, 1T-IrO₂ shows the high stability for the start-stop cycles test.

Supplementary Figure 25. a, The start-stop cycles stability test of 1T-IrO₂ and **b**, the start-stop cycles stability test in the first one hour.

[Added Content]: “Supplementary Table 9. The comparison of stability performances of 1T-IrO₂ and the reported electrocatalysts in acidic OER.

Catalyst	Electrolyte	Current Density (mA cm _{geo} ⁻²)	Time (h)	Ref.
1T-IrO ₂	0.1 M O ₂ -saturated HClO ₄	50	45	This work
1T-IrO ₂	0.5 M O ₂ -saturated H ₂ SO ₄	40	18	This work
1T-IrO ₂	0.5 M O ₂ -saturated H ₂ SO ₄ in PEM	250	126	This work
Sr ₂ CoIrO ₆	0.1 M HClO ₄	10	24	21
Ir-MnO ₂	0.5 M H ₂ SO ₄	10	650	22
Mesoporous Ir nanosheets	0.5 M H ₂ SO ₄	10	8	23

21. Zhang, R. H. et al. A dissolution/precipitation equilibrium on the surface of iridium-based perovskites controls their activity as oxygen evolution reaction catalysts in acidic media. *Angew. Chem. Int. Ed.* **58**, 4571-4575 (2019).

22. Shi, Z. P. et al. Confined Ir single sites with triggered lattice oxygen redox: toward boosted and sustained water oxidation catalysis. *Joule* **5**, 2164-2176 (2021).

23. Jiang, B. et al. Mesoporous metallic iridium nanosheets. *J. Am. Chem. Soc.* **140**, 12434-12441 (2018).”

[Page 11, Line 15] “As shown in **Supplementary Fig. 25**, 1T-IrO₂ shows the high stability for the start-stop cycles test.”

5. The stability test (p. 10) is only carried out with 1T-IrO₂ and C-IrO₂ but not with the other

reference materials. This should be changed and the other materials (Rutile, C-Ir/C) should be included and compared.

[Author's Response]: Thank you for your comment. According to your suggestions, we have provided the data by carrying out stability tests of Rutile-IrO₂ and C-Ir/C in **Supplementary Fig. 16**. The stability tests were carried out by chronopotentiometry method under a constant current density of 25 mA cm_{geo}⁻². As shown in **Supplementary Fig. 16**, the activities of Rutile-IrO₂ and C-Ir/C decrease sharply, respectively.

Supplementary Figure 16. Stability tests for Rutile-IrO₂ and C-Ir/C. Chronopotentiometry performances of **a**, Rutile-IrO₂ and **b**, C-Ir/C under a constant current density of 25 mA cm_{geo}⁻².

[Added Content]: [Page 10, Line 15] “However, Rutile-IrO₂, C-IrO₂ and C-Ir/C totally deactivated after the long-term test under the high current density (**Fig. 3g and Supplementary Fig. 16**).”

6. The computational investigation on the structure of the 1T-IrO₂ (page 6, second paragraph) is sound but the authors should express some words of caution - just because a structure is a local minimum, it does not mean it is stable in solution or at higher potential. Furthermore, structures have only been optimized for the bulk, as far as I understand. As for catalysis the interface is essential, DFT calculations should also focus on surface stability for different surface terminations (ab-initio thermodynamics / computational Pourbaix diagrams, see work by Scheffler, Gross, Todorova etc.).

[Author's Response]: The calculations in **Page 6, Line 14** was adopted to confirm that the T-phase oxygen coordination is a local minimum configuration and is more energetic favorable than the H-phase. The stability of 1T-IrO₂ in solution or at higher potential mainly concluded from the experimental characterization. We revised this expression to make it more rigorous. Meanwhile, based on the reviewer's suggestion, we also plotted the computational Pourbaix diagram of 1T-IrO₂ (0001) surface and took the effect of different surface terminations into consideration. As shown in **Supplementary Fig. 27**, it was found that at U < 1.76 V/RHE, the bare 1T-IrO₂ (0001) surface had the lowest formation energy, which was the dominant reaction surface for the OER.

Supplementary Figure 27. Computational Pourbaix diagram of 1T-IrO₂(0001) surface. All surface formation energies at different coverage were referenced to that of the bare surface.

[Added Context]: [Page 6, Line 14] “The H phase structure of IrO₂ collapsed upon structure relaxation while the T phase of IrO₂ keeps stable, suggesting that T-phase is more energetic-favorable at ground state.”

[Page 13, Line 1] “Computational Pourbaix diagram indicates the reaction surface of 1T-IrO₂ mainly happens on the bare (0001) surface at $U < 1.76$ V/RHE (**Supplementary Fig. 27**) and the active site has been assigned to the Ir atoms.”

7. The computational study of the "reactivity" of the material (Fig. 4) apply conventional DFT methods and an a-posteriori approach for the inclusion of the electrochemical potential. While the authors did choose to include implicit solvent and the results are consistent and allow for some degree of comparison, this approach is not really state of the art anymore. Simply computing adsorption energies of molecular and atomic species is a rough indicator, but the electronic structure of the material is undefined in terms of the electrochemical potential. Modern investigations using electronic structure theory for OER and ORR should explicitly include the effect of the electrochemical potential in the simulations (Work by Goddard, Neugebauer, Bhattacharyya etc.) and give a more accurate picture of structures, energetics and reactivity. In order to live up to the high standards of Nature Communications, the authors should adapt a more accurate description of the underlying physics in their models. This way, the simulation would also yield more details on how the material behaves and how reactivity is influenced by the potential. Currently, all values are energies of neutral systems including implicit solvation, which are scaled a-posteriori for a rough estimate of the potential influence.

[Author's Response]: Based on the reviewer's comment, we adopted JDFTx software to calculate the thermodynamic free energy change of 1T- and rutile-IrO₂ at the potential-determining step and considered the effect of the electrode potential and the solvation. Similar with the computational details in the work by Yuan Ping et. al. (Ping, Y. et al. The reaction mechanism with free energy barriers at constant potentials for the oxygen evolution reaction at the IrO₂ (110) surface. J. Am. Chem. Soc. 139, 149-155 (2017)), the constant potential model along with the CANDLE implicit solvation model was set to compute the grand free energy. We considered the frequency vibration correction of the Ir atom and calculated the limiting overpotentials on rutile-IrO₂ (110) and 1T-IrO₂ (0001) surfaces are corresponding to 0.59 V at the step of *O → *OOH and 0.52 V at the step of * → *OH, respectively. In the constant potential model (JDFTx), it was found that the limiting overpotential of 1T-IrO₂ (0001) decreased to 0.41 V while that of rutile-IrO₂ (110) increased to 0.75 V (**Supplementary Table 13**). Therefore, both of the constant-charge and constant-potential simulation results predicted that 1T-IrO₂ has a better OER performance than rutile phases. Accordingly, we added more discussions in the computational details.

Supplementary Table 13. Comparison of the limiting overpotentials on rutile IrO₂(110) and 1T IrO₂(0001) by the constant charge model and the constant potential model.

Limiting Overpotential (V/RHE)	1T-IrO ₂ (0001) (eV)	Rutile-IrO ₂ (110) (eV)
Constant Charge Model (VASP)	0.52	0.59
Constant Potential Model (JDFTx)	0.41	0.75

[Added Context]: [Page 20, Line 10] “Considering the effect of the electrochemical electrode potential and the solvation, we also used JDFTx software⁶⁰ to calculate the U_{limiting} at the

constant electrode potential of 1.23 V along with the CANDLE implicit solvation model⁶¹ and found that both of the constant-charge and constant-potential simulation results have the same trend. The quantitative effects of electrode potential and functionals are given in **Supplementary Tables 13, 14**.

60. Sundararaman, R., Letchworth-Weaver, K., Schwarz, K. A., Gunceler, D., Ozhabes, Y. & Arias, T. A. JDFTx: Software for joint density-functional theory. *SoftwareX* **6**, 278-284 (2017).

61. Sundararaman, R. & William A. Goddard III, The charge-asymmetric nonlocally determined local-electric (CANDLE) solvation model. *J. Chem. Phys.* **142**, 064107 (2015).”

8. While computational details are given in the manuscript and one figure on the computed structures is given in the SI, no further details are provided. All simulated structures should be given in the SI for reproducibility. Furthermore, the authors have used one Functional for all calculations - here it would be desirable to also apply other functionals and report differences so that the reader can assess deviations.

[Author’s Response]: According to the reviewer’s suggestion, we provided all simulated geometry inputs in **Supplementary Note 2** for reproducibility and compared the U_{limiting} calculated from four functionals (PBE, PBE+D3, RPBE, revPBE and PW91) in **Supplementary Table 14**.

Supplementary Table 14. Theoretical overpotentials based on the PBE, PBE-D3, RPBE, revPBE and PW91 functionals.

1T-IrO ₂	PBE	PBE-D3	RPBE	revPBE	PW91
Overpotentials (V)	0.54	0.52	0.63	0.62	0.46

[Added Content]: [Page 20, Line 14] “The quantitative effects of electrode potential and functionals on the U_{limiting} were given in **Supplementary Tables 13, 14**.”

REVIEWERS' COMMENTS

Reviewer #2 (Remarks to the Author):

The authors added all the experimental details requested and clarified other points and sentences according to my suggestions. I recommend the work for publication in Nature Communications.

Reviewer #3 (Remarks to the Author):

The authors have made a considerable effort to improve on the quality of the manuscript and the amount of data supporting their claims and statements. Novel results have been added and especially open questions concerning the electronic structure simulations have been answered thoroughly including further calculations with different models. Hence, in my view, the paper can now be published in Nature Communications.

Response to Reviewers

Dear Reviewers,

Thank you for your precious time to constructive comments on our manuscript titled “Iridium Metallene Oxide for Acidic Oxygen Evolution Catalysis” (Manuscript ID: NCOMMS-21-26540-A) for Nature Communications.

Reviewer #2

The authors added all the experimental details requested and clarified other points and sentences according to my suggestions. I recommend the work for publication in Nature Communications. **[Author’s Response]:** We would like to thank you for the positive reports and recommendation of the publication in the Nature Communications.

Reviewer #3

The authors have made a considerable effort to improve on the quality of the manuscript and the amount of data supporting their claims and statements. Novel results have been added and especially open questions concerning the electronic structure simulations have been answered thoroughly including further calculations with different models. Hence, in my view, the paper can now be published in Nature Communications. **[Author’s Response]:** Thank you very much for your recommendation of the publication.